# Cryo-EM structures of CTP synthase filaments reveal mechanism of pH-sensitive assembly during budding yeast starvation

Jesse M Hansen[1,2†], Avital Horowitz[1†], Eric M Lynch[1], Daniel P Farrell[1], Joel Quispe[1], Frank DiMaio[1], Justin M Kollman[1]*

[1]Department of Biochemistry, University of Washington, Seattle, United States; [2]Graduate Program in Biological Physics, Structure, and Design, University of Washington, Seattle, United States

**ABSTRACT** Many metabolic enzymes self-assemble into micron-scale filaments to organize and regulate metabolism. The appearance of these assemblies often coincides with large metabolic changes as in development, cancer, and stress. Yeast undergo cytoplasmic acidification upon starvation, triggering the assembly of many metabolic enzymes into filaments. However, it is unclear how these filaments assemble at the molecular level and what their role is in the yeast starvation response. CTP Synthase (CTPS) assembles into metabolic filaments across many species. Here, we characterize in vitro polymerization and investigate in vivo consequences of CTPS assembly in yeast. Cryo-EM structures reveal a pH-sensitive assembly mechanism and highly ordered filament bundles that stabilize an inactive state of the enzyme, features unique to yeast CTPS. Disruption of filaments in cells with non-assembly or pH-insensitive mutations decreases growth rate, reflecting the importance of regulated CTPS filament assembly in homeotstasis.

*For correspondence:
jkoll@uw.edu

#These authors contributed equally.

## Editor's evaluation

This work provides valuable new information to those who study enzyme mechanisms, nucleotide metabolism, and the response of cells to stress such as nutrient deprivation. The study focuses on CTP Synthase (CTPS), an important enzyme in nucleotide biosynthesis that has been shown to assemble into foci and filaments in yeast cells undergoing starvation conditions. The authors study the structure of yeast CTPS and its propensity to polymerize in low pH (mimicking starvation conditions), and how CTPS filamentation relates to the cellular assemblies.

## Introduction

Intermediate metabolism is finely tuned, carefully balanced, and robustly adaptable to changes in environmental conditions. More than a century of effort has gone into understanding the connections between pathways, the individual enzymes that drive the biochemistry, and the regulation of metabolic flux. Only recently, however, have we come to appreciate the role of metabolic enzyme self-assembly as a widespread mechanism of metabolic organization and regulation (*Lynch et al., 2020*; *Park and Horton, 2019*; *Simonet et al., 2020*). These assemblies have been found in many core pathways including glycolysis (*Kemp, 1971*; *Webb et al., 2017*), fatty acid synthesis (*Hunkeler et al., 2018*; *Kleinschmidt et al., 1969*), amino acid synthesis (*Cohen et al., 1976*; *Frey et al., 1975*; *Miller et al., 1974*; *Petrovska et al., 2014*; *Zhang et al., 2018*), and nucleotide synthesis (*Barry*

**Figure 1.** CTPS canonical structure and filament assembly. (**a**) De novo CTP synthesis reaction diagram. (**b**) Substrate-bound hCTPS2 (6PK4) monomer with glutamine modeled from 1VCO. (**c**) Canonical tetramer structure of CTPS (6PK4). Monomers are shaded differently, and domain colors from panel B are painted for one monomer. (**d**) Surface model display for filament assembly of stacked tetramers for product-bound *E. coli* CTPS (5U3C). (**e**) Surface model display for filament assembly of stacked tetramers for product-bound hCTPS2 (6PK7). (**f**) Zoom-in of box from panel E showing filament assembly interface, with conserved H355 in red. Below, sequence alignment of filament assembly interface, with blue shaded box for eukaryotic alpha helical insert and conserved key histidine in red.

*et al., 2014*; *Carcamo et al., 2011*). Metabolic enzymes that form filaments are frequently found at rate-limiting and energetically committed steps of pathways, suggesting filaments play a role in regulating metabolic flux (*Noree et al., 2019*). Indeed, for most enzyme polymers that have been functionally characterized, assembly functions as a mechanism of allosteric regulation to tune enzyme activity (*Barry et al., 2014*; *Hunkeler et al., 2018*; *Johnson and Kollman, 2020*; *Lynch et al., 2017*; *Lynch and Kollman, 2020*; *Stoddard et al., 2020*).

CTP synthase (CTPS) catalyzes the final, rate-limiting step of de novo CTP biosynthesis (*Lieberman, 1956*; *Figure 1a*). Its biochemistry and structure have been studied extensively, making it an ideal model enzyme for exploring the role of enzyme assembly in regulating activity. Each CTPS monomer consists of a glutaminase domain and an amido-ligase domain connected by an alpha helical linker (*Figure 1b*). Glutamine hydrolysis in the glutaminase domain produces ammonia which is transferred through a channel into the amido-ligase domain, where it is ligated to UTP to form CTP in an ATP-dependent process (*Endrizzi et al., 2005*; *Goto et al., 2004*). Substrate binding induces rotation of the glutaminase domain which opens an ammonia channel between active sites (*Lynch and Kollman, 2020*), and it's reaction product CTP inhibits activity (*Habrian et al., 2016*). CTPS assembles X-shaped, D2 symmetric homotetramers through interactions of the amido-ligase domains, with multiple protomers participating in each substrate and regulatory binding site (*Figure 1c*).

Cellular CTPS filament assembly is widespread across the domains of life, having been observed in bacteria, archaea, and eukaryotes (*Carcamo et al., 2011*; *Ingerson-Mahar et al., 2010*; *Liu, 2010*; *Noree et al., 2010*; *Zhou et al., 2021*). We have previously shown, however, that there are striking differences between the assembly mechanisms and filament architectures among species

(*Figure 1d–f*). These differences give rise to differences in function, with bacterial CTPS filaments providing a mechanism to allosterically inhibit the enzyme, while animal CTPS filaments act to increase activity and cooperativity of regulation (*Barry et al., 2014*; *Lynch et al., 2017*; *Lynch and Kollman, 2020*; *Zhou et al., 2021*; *Zhou et al., 2019*).

In budding yeast, regulation of metabolism is key to survival as sudden fluctuations in environmental resources is common. During starvation, yeast assemble many metabolic enzymes into filaments (*Narayanaswamy et al., 2009*; *Noree et al., 2010*; *Shen et al., 2016*) which protects cells and permits rapid growth upon readdition of nutrients (*Petrovska et al., 2014*). It is thought that polymerization leads to increased cytoplasmic viscosity, limiting diffusion of metabolites and slowing growth (*Petrovska et al., 2014*). Without energy, membrane-bound proton pumps fail and the neutral-pH cytoplasm is acidified (*Orij et al., 2009*). This acidification is necessary and sufficient for yeast to trigger metabolic filament assembly and effectively mount a stress response (*Petrovska et al., 2014*). Therefore, yeast present a unique circumstance to study metabolic filament assembly as it relates to the stress response. Fluorescence-based studies of tagged CTPS in yeast and other organisms demonstrated large foci or rods consistent with laterally associating filaments (*Ingerson-Mahar et al., 2010*; *Liu, 2010*), yet there are no high-resolution structures of bundled metabolic enzymes in any organism. The molecular basis of starvation-driven assembly remains unknown as well as whether polymer assembly regulates enzymatic activity of CTPS during the yeast stress response.

Budding yeast have two CTPS isoforms, Ura7 (*Yang et al., 1994*) and Ura8 (*Nadkarni et al., 1995*; *Ozier-Kalogeropoulos et al., 1994*), which share 78 % identical residues. While Ura7 mRNA is twofold more abundant than Ura8 in vivo (*Nadkarni et al., 1995*), they are functionally overlapping and deletion of either gene results in slowed growth (*Ozier-Kalogeropoulos et al., 1994*). Both form supramolecular structures in response to stress that colocalize in cells (*Noree et al., 2010*; *Shen et al., 2016*).

Here, we investigate the structure and in vivo function of yeast CTPS filaments. We use cryo-electron microscopy (cryo-EM) to determine the molecular mechanism of yeast CTPS filament assembly and higher-order bundled assemblies. The interface between tetramers in the yeast CTPS filament, which is unique among the existing CTPS filament structures, explains the pH sensitivity of assembly. Two engineered mutations at this assembly interface, one that disrupts filaments and one that stabilizes them, allow us to probe the functional role of filament assembly in vitro and in vivo. Yeast CTPS filaments stabilized a conformation that pinches shut the ammonia channel between catalytic sites, which reduces activity in the filament. Both non-assembling and pH-insensitive CTPS mutants lead to slowed proliferation, indicating the critical role of regulated CTPS assembly in both vegetative growth and recovery from starvation.

## Results
### Yeast CTPS filament assembly is pH-sensitive

It remains an open question whether pH-regulated assembly is an inherent feature of budding yeast CTPS, or whether other cellular factors are required for assembly. To address this, we first confirmed prior work that showed yeast CTPS assembles in cells upon cytoplasmic acidification (*Petrovska et al., 2014*); Ura7-GFP tagged at the endogenous locus forms foci upon nutrient deprivation (*Figure 2a*), or when cells are permeabilized with DNP (2,4-dinitrophenol) to manipulate cytoplasmic pH (*Figure 2b*). We next examined polymerization of purified recombinant Ura7 and Ura8 at different pHs by negative stain EM. Apo CTPS did not form polymers at any pH. At pH 7.4 both isoforms assembled short, single filaments on binding substrates or product, but pH 6.0 promoted assembly of much larger polymers that appeared to be bundled filaments (*Figure 2c*, *Figure 2—figure supplement 1*). Thus, like other species (*Barry et al., 2014*; *Lynch et al., 2017*; *Zhou et al., 2019*), yeast CTPS assembly is dependent on ligand binding, and the pH sensitivity is intrinsic to the enzyme itself.

We wondered whether the intrinsic pH sensitivity of yeast CTPS was conserved among other species. Previous studies have not described the pH-sensitivity of human CTPS filament assembly, and have shown robust assembly at pH 8.0 (*Lynch et al., 2017*; *Lynch and Kollman, 2020*). The overall sequence conservation between human and yeast CTPS, and the presence of a histidine residue at the assembly interface in human CTPS filaments, led us to consider whether human CTPS assembly is also pH-dependent (*Figure 1d*). We observed equally robust polymerization of purified human CTPS2 (hCTPS2) at pH 6.0 and 8.0, and negative stain EM averages show that filaments formed at different

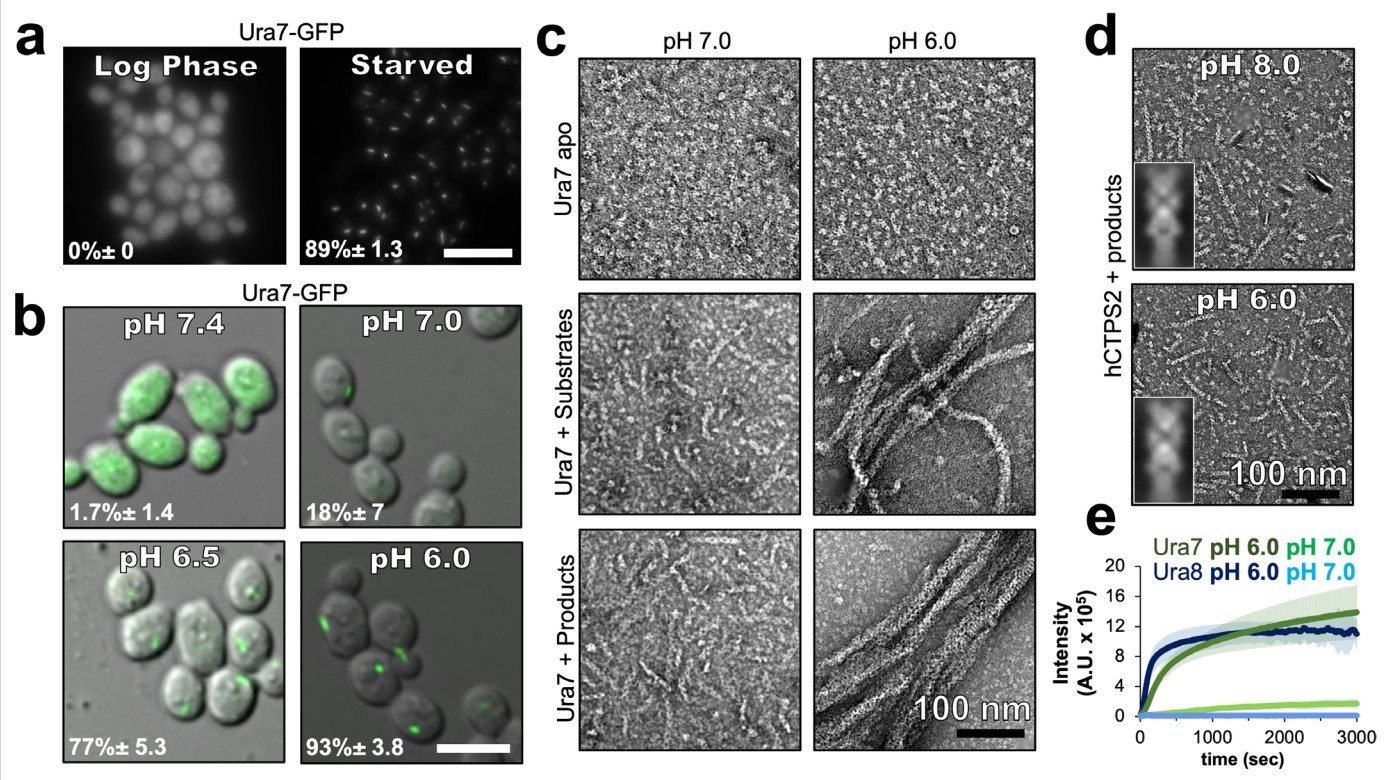

**Figure 2.** Yeast CTPS assembly is driven by pH with addition of substrates or products. (**a**) Yeast expressing GFP-tagged Ura7 in log phase and starvation media. Quantification is shown as a percentage of cells with foci. Scale bar 10 μm. (**b**) Yeast expressing GFP-tagged Ura7 with cell membrane permeabilized using 2 mM DNP supplemented with 2 % glucose. Quantification is shown as a percentage of cells with foci. Scale bar 10 μm. (**c**) Negative stain EM of purified Ura7 assembled with substrate (2 mM UTP/ATP) or products (2 mM CTP). (**d**) Negative stain EM of purified hCTPS2 assembled with 2 mM CTP. Insets are representative 2D class averages. (**e**) Right angle light scattering with addition of 2 mM CTP. Units of intensity are in arbitrary units (A.U.).

The online version of this article includes the following figure supplement(s) for figure 2:

**Figure supplement 1.** Ura8 assembly is driven by pH and substrates or products.

**Figure supplement 2.** Right angle light scattering assembly kinetics for CTPS filaments.

pHs have similar architectures, suggesting that there is no direct effect of pH on hCTPS2 assembly or structure (*Figure 2d*, inserts). Thus, pH-driven assembly of CTPS appears to be specific to yeast, raising the question of what unique structural features might confer pH-sensitivity.

To assess the kinetics of pH-dependent CTPS assembly, we monitored right angle light scattering by CTPS after addition of CTP (*Figure 2e*). Both isoforms had very low signal at pH 7, consistent with our negative stain imaging. At pH 6, Ura8 assembly was much faster than Ura7, but inspection at early time points showed that both exhibited biphasic assembly which could not be fit with a single equation (*Figure 2—figure supplement 2*). Instead, early and late assembly kinetics fit well to separate four-parameter curves, indicating potentially distinct assembly phenomena with different kinetics in the early and late phases. We examined the growth of Ura7 filaments by negative stain EM over time, and found that single filaments appear at early time points and the thicker bundles at later time points, suggesting that the biphasic scattering kinetics can be explained by initial linear polymerization followed by lateral aggregation (*Figure 2—figure supplement 2*).

## Structures of yeast CTPS filaments

We next determined structures of individual CTPS filaments by cryo-EM. To enable structure determination, we assembled filaments at pH 6.5, where single filaments predominate over larger bundles. We solved filament structures of both Ura7 and Ura8 in substrate- (ATP and UTP) and product- (CTP) bound states. Ura8 yielded the highest resolution structures, at 2.8 Å (substrates) and 3.8 Å (products)

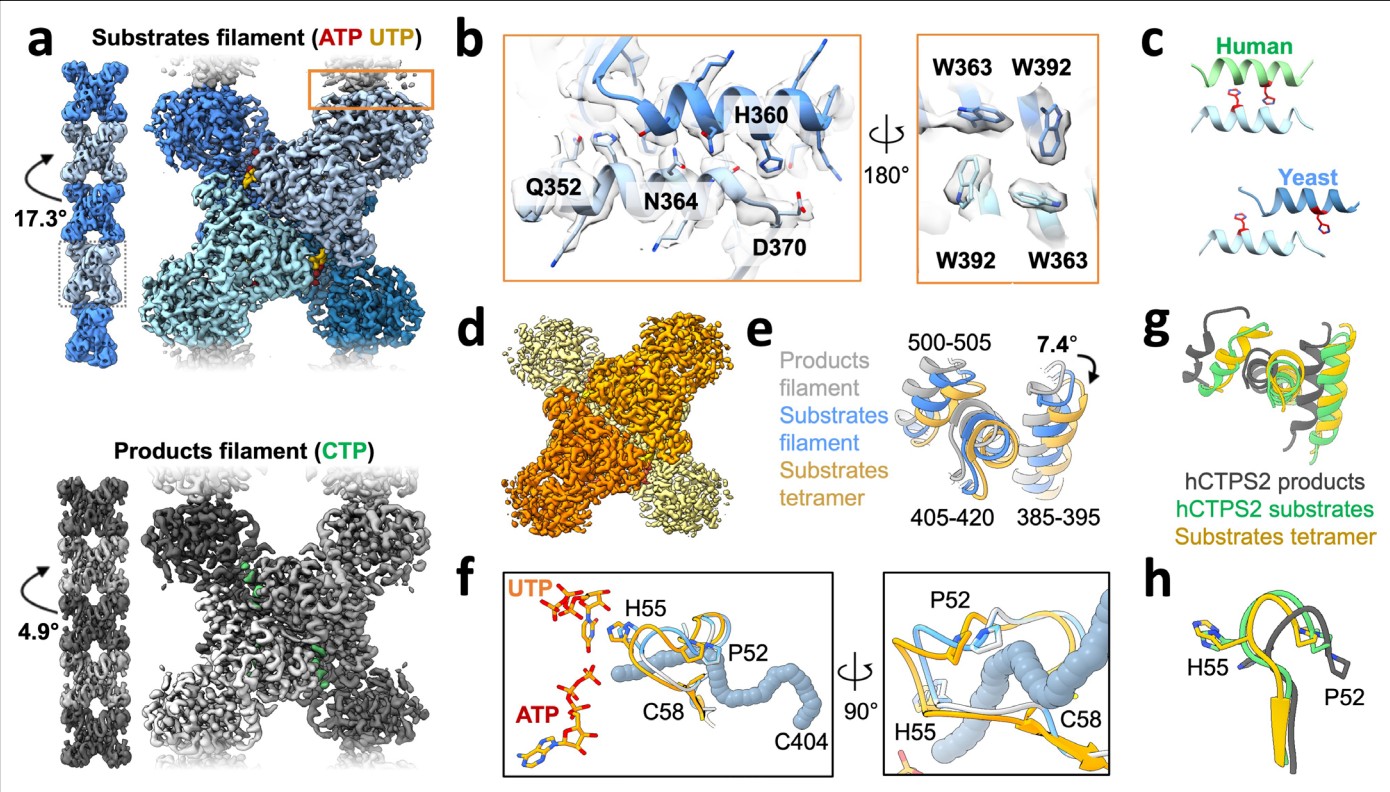

**Figure 3.** Yeast CTPS assembles distinct filaments which are not in the canonical active conformation. (**a**) Cryo-EM maps of Ura8 filaments assembled at pH 6.0 in the presence of substrates (top; UTP and ATP) or product (bottom; CTP). Left, filament map from imposing helical symmetry parameters on a reconstruction of a single protomer, and low-pass filtered to 15 Å. Dotted box delineates an individual tetramer. (**b**) Zoom-in of the filament assembly interface (orange box in panel A) with superimposed cryo-EM density. Key residues are indicated. Right panel is the back-side view of the interface. (**c**) Comparison of human and yeast interface, with H360 (H355 in humans) highlighted in red. (**d**) Cryo-EM map of substrate-bound (ATP,UTP) tetramer at pH 7.4. (**e**) Glutaminase domain rotation (residues indicated) of monomers aligned on the amido-ligase domain. (**f**) Substrate-channel (gray tube) beginning at catalytic C404 ending at ligands UTP/ATP. Color scheme same as panel E. (**g**) Ura8 substrate-bound tetramer glutaminase domain rotation relative to hCTPS2 substrate (6PK4) and product-bound (6PK7) filaments. (**h**) Loop movements near the active site which alleviates constriction in the substrate-channel. Color scheme same as panel G.

The online version of this article includes the following video and figure supplement(s) for figure 3:

**Figure supplement 1.** Nucleotides in yeast CTPS.

**Figure supplement 2.** Ura7 filament architecture and assembly interface.

**Figure supplement 3.** Protonation state at the filament assembly interface.

**Figure supplement 4.** Comparison of buried surface area for yeast and human CTPS.

**Figure supplement 5.** Amido-ligase domain movement at the tetramerization interface.

**Figure supplement 6.** CTPS filaments are incompatible with the active state.

**Figure supplement 7.** Image processing of substrate-bound Ura8 filaments.

**Figure supplement 8.** Image processing of product-bound Ura8 filaments.

**Figure supplement 9.** Image processing of substrate-bound Ura8 tetramers.

**Figure supplement 10.** Image processing of substrate-bound Ura7 filaments.

**Figure supplement 11.** Image processing of product-bound Ura7 filaments.

**Figure 3—video 1.** Ura8 conformational changes.

https://elifesciences.org/articles/73368/figures#fig3video1

(*Figure 3a*, *Table 1*). In the Ura8 structures, all ligands were clearly visible and bound as previously described for other CTPS homologs, including an imino-phosphate reaction intermediate recently observed in *Drosophila* CTPS, and the presence of two distinct CTP-binding sites per monomer recently reported in human and *Drosophila* CTPS (*Endrizzi et al., 2004*; *Goto et al., 2004*; *Lynch*

**Table 1.** Cryo-EM data collection and refinement statistics.

| | Ura8 substrates filament | Ura8 substrates bundle | Ura8 substrates tetramer | Ura8 products filament | Ura8 products bundle | Ura7 substrates filament | Ura7 substrates bundle | Ura7 products filament | Ura7 products bundle | Ura7-H360R substrates filament |
|---|---|---|---|---|---|---|---|---|---|---|
| Number of micrographs | 2729 | 2729 | 1464 | 4979 | 1921 | 2819 | 2819 | 3118 | 2181 | 1070 |
| Nominal magnification | 130,000 X | | | | | | | | | |
| Voltage | 300 kV | | | | | | | | | |
| Electron Fluence | 90e-/Å² | | | | | | | | | |
| Pixel size | 1.05 Å | | | | | | | | | |
| Defocus range | -0.4 to -1.9 μm | -0.4 to -1.9 μm | -1.2 to -5.5 μm | -0.4 to -9.0 μm | -0.4 to -7.0 μm | -0.8 to -2.2 μm | -0.8 to -2.2 μm | -0.4 to -7.0 μm | -1.0 to -2.2 μm | -0.9 to -3.5 μm |
| EMDB ID | EMD-24512 | EMD-24581 | EMD-24497 | EMD-24516 | EMD-24579 | EMD-24566 | EMD-24575 | EMD-24560 | EMD-24576 | EMD-24578 |
| PDB code | 7RL0 | 7RNR | 7RKH | 7RL5 | 7RNL | 7RMF | 7RMK | 7RMC | 7RMO | 7RMV |
| Map resolution 0.143 FSC | 2.9 Å | 3.4 Å | 2.9 Å | 4.0 Å | 4.1 Å | 7.3 Å | 6.6 Å | 3.7 Å | 7.0 Å | 6.7 Å |
| Density Modified Resolution 0.5Ref | 2.8 Å | 3.3 Å | 2.8 Å | 3.8 Å | 3.7 Å | n/a | n/a | 3.5 Å | n/a | n/a |
| Symmetry Imposed | D2 | C2 | D2 | D2 | C2 | D2 | C2 | D2 | C2 | D2 |
| Number of particles | 40,474 | 21,220 | 76,963 | 181,136 | 53,445 | 19,706 | 24,059 | 64,010 | 28,214 | 54,928 |
| Map Sharpening B Factor | -32 Å² | -33 Å² | -74 Å² | -142 Å² | -67 Å² | -235 Å² | -140 Å² | -51 Å² | -186 Å² | -40 Å² |
| **Model composition** | | | | | | | | | | |
| ligands | ATP/UTP | ATP/UTP | ATP/UTP | CTP | CTP | ATP/UTP | ATP/UTP | CTP | CTP | ATP/UTP |
| **R.M.S. deviations** | | | | | | | | | | |
| Bond lengths (Å) | 0.64 | 0.63 | 0.63 | 0.66 | 0.7 | 0.69 | 0.69 | 0.62 | 0.63 | 0.71 |
| Bond angles (°) | 1.09 | 1.04 | 1.04 | 1.09 | 1.13 | 1.06 | 1.06 | 1.03 | 1.03 | 1.07 |
| **Validation** | | | | | | | | | | |
| Molprobity Score | 0.98 | 1.35 | 1.09 | 1.59 | 1.66 | 1.56 | 2.06 | 1.40 | 1.73 | 2.30 |

*Table 1 continued on next page*

*Table 1 continued*

| | Ura8 substrates filament | Ura8 substrates bundle | Ura8 substrates tetramer | Ura8 products filament | Ura8 products bundle | Ura7 substrates filament | Ura7 substrates bundle | Ura7 products filament | Ura7 products bundle | Ura7-H360R substrates filament |
|---|---|---|---|---|---|---|---|---|---|---|
| Clash Score | 0.94 | 4.74 | 0.9 | 3.2 | 3.48 | 2.31 | 10.33 | 1.51 | 4.62 | 15.84 |
| Poor Rotamers (%) | 1.21 | 0.61 | 1.64 | 2.05 | 1.43 | 1.88 | 1.88 | 1.65 | 1.66 | 2.09 |
| **Ramachandran Plot** | | | | | | | | | | |
| Favored (%) | 97 | 98 | 97 | 96 | 94 | 95 | 95 | 95 | 95 | 95 |
| Allowed (%) | 2 | 2 | 3 | 3 | 6 | 4 | 4 | 5 | 4 | 5 |
| Outliers (%) | 0 | 0 | 0 | 0 | 0 | 0 | 0 | 0 | 0 | 0 |

**Table 2.** CTPS polymer architecture characteristics.

| | Ura8 prods filament | Ura8 subs filament | Ura8 subs tetramer | Ura8 prods bundle | Ura8 subs bundle | Ura7 prods filament | Ura7 subs filament | Ura7 prods bundle | Ura7 subs bundle |
|---|---|---|---|---|---|---|---|---|---|
| Domain rotation relative to product-bound filament state | n/a | 4° | 7.4° | 1.7° | 6.9° | n/a | 7.7° | 1.4° | 6.2° |
| Filament twist | 5° | 17.1° | n/a | 3.9° | 15.9° | 7.4° | 18.2° | 7.3° | 17.3° |
| Filament rise | 102.6 Å | 102.3 Å | n/a | 103 Å | 101.8 Å | 102.7 Å | 103.4 Å | 102.8 Å | 104.5 Å |

*et al., 2021*; *Lynch et al., 2017*; *Lynch and Kollman, 2020*; *Zhou et al., 2021*; *Figure 3—figure supplement 1c*). Ura7 filaments reached lower resolutions, 7.3 Å (substrates) and 3.7 Å (products), but the overall structures were indistinguishable from Ura8 at these resolutions (*Figure 3—figure supplement 2*, *Table 2*). Because of their higher resolution which enabled building of atomic models, we focus subsequent structural interpretation on the Ura8 filaments.

Like other previously reported structures of human, *Drosophila*, and *E. coli* CTPS, the yeast enzyme assembles as stacked tetramers. However, it does so using a completely different interaction interface, which was the same in both substrate- and product-bound yeast structures (*Figure 3b*). Helix 356–370 in the glutaminase domain mediates interactions between tetramers, so that each protomer is involved in assembly contacts (*Figure 3b*). His360 interacts with D370 which would be stabilized by protonation of His360 at low pH, and likely explains the pH-sensitivity of the assembly interaction (*Figure 3b*, *Figure 3—figure supplement 3*). We see density consistent with two rotamer positions of H360 that may reflect partial protonation of the sidechain at pH 6.5. One rotamer that likely reflects the protonated state interacts with D370, and the other rotamer that likely reflects an unprotonated state forms a hydrogen bond to a backbone carbonyl across the interface (*Figure 3—figure supplement 3a–c*). Additional contacts include hydrogen bonding between an asparagine pair on the two-fold symmetry axis (Asn364), hydrogen bonding (Gln352) to a backbone carbonyl across the interface, hydrophobic interactions of a cluster of tryptophans (Trp363, Trp392), and a pair of salt bridges (Lys391, Glu395) (*Figure 3b*, *Figure 3—figure supplement 2*). Helix 356–370 also forms the assembly interface of animal CTPS filaments, but the yeast interface is shifted by two turns of the helix, resulting in completely different residue contacts (*Figure 3c*). This leads to a larger interaction interface in yeast CTPS filaments (706 Å$^2$ per monomer in product-bound filaments) than in the human homolog (492 Å$^2$ per monomer in product-bound filaments) (*Figure 3—figure supplement 4*). Sequence differences between human and yeast CTPS at residues 364, 391, and 395 may explain how the unique yeast interface arose, which shifted H360 into a position to mediate a pH-sensitive interaction.

The primary difference between the substrate- and product-bound filament structures is the conformation of individual CTPS protomers. Active and inactive conformations of CTPS are characterized by a ~ 7° rotation of the glutaminase domain relative to the amido-ligase domain, which opens an ammonia channel between the two active sites in a single protomer (*Lynch and Kollman, 2020*). The Ura8 product-bound filament is in the canonical inhibited conformation, but the substrate-bound structure is in a conformation intermediate between canonical active and inhibited states, in which the ammonia channel remains closed. This suggested to us that one function of yeast CTPS filaments is to constrain the enzyme in a low activity conformation. To test this hypothesis, we determined the structure of free Ura8 tetramers at pH 7.4 bound to substrates at 2.8 Å resolution. In this unassembled state, Ura8 adopts the canonical active conformation, very similar to hCTPS2, including opening of the ammonia channel and rearrangement at the tetramerization interface (*Figure 3d–h*, *Figure 3—figure supplement 5*, *Figure 3—video 1*).

The active state we observe in free tetramers is incompatible with filament assembly, because in this conformation filament interfaces cannot be occupied on both sides of the tetramer simultaneously (*Figure 3—figure supplement 6*). Indeed, the major source of heterogeneity in single Ura7-substrate filaments, which were flexible and short, are tetramers that only make one contact at each paired assembly interface. This leads to tetramers in the active, substrate-bound conformation tethered to each other through single glutaminase domain interactions (*Figure 3—figure supplement*

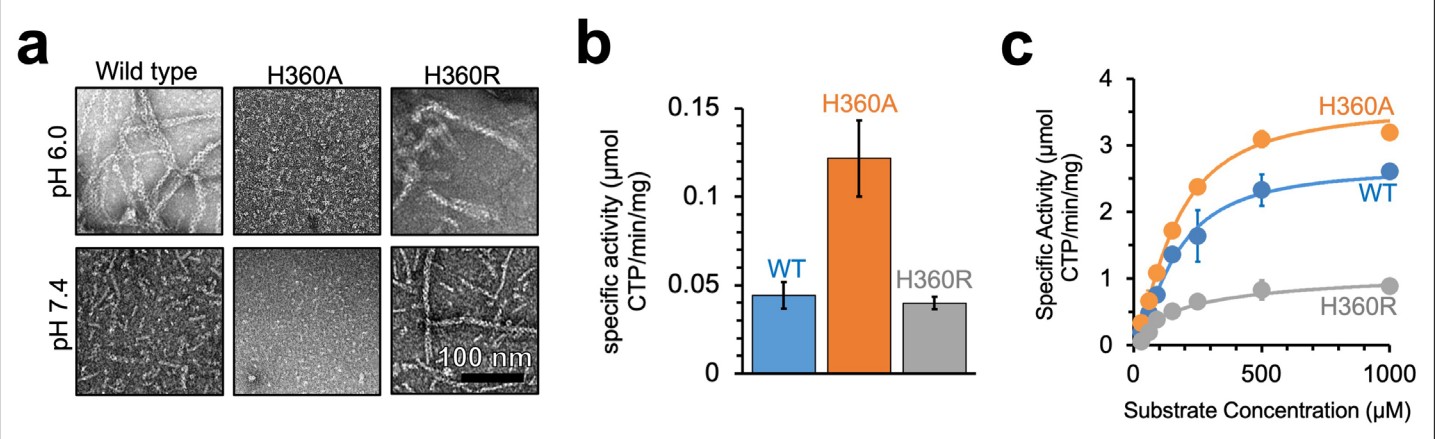

**Figure 4.** Yeast CTPS polymers reduce enzymatic activity. (**a**) Negative stain EM of purified wild type and mutant Ura7 with addition of 2 mM CTP. (**b**) Activity assay with saturating substrates of Ura7 and mutants at pH 6.0. (**c**) Substrate kinetics of wild type and mutant Ura7 at pH 7.4 with four parameter regression curve fits. Experiments done in triplicate, error bars are standard error of the mean (SEM).

The online version of this article includes the following figure supplement(s) for figure 4:

**Figure supplement 1.** Ura7 H360R ligand-dependent assembly.

**Figure supplement 2.** Ura7 H360R structural validation.

**Figure supplement 3.** Image processing of substrate-bound Ura7-H360R at pH 7.5.

*6d*). Thus, filament assembly with symmetric contacts acts as a steric constraint to trap yeast CTPS in a low activity conformation, while tethering through a single glutaminase domain supports short, flexible filaments that accommodate a fully active conformation.

## Yeast CTPS filaments reduce enzyme activity

To test whether CTPS filament assembly has a direct effect on enzyme activity, we generated mutations at H360 in the assembly interface of Ura7 that either disrupted or stabilized filaments. First, we generated H360A, which disrupts polymerization of the human enzymes, and which our structural analysis suggests is the pH sensor in assembly of the yeast enzyme (*Lynch et al., 2017*; *Lynch and Kollman, 2020*). Consistent with those prior results, Ura7-H360A did not assembly filaments with substrates or products at low or high pH (*Figure 4a*). To mimic protonation at H360, we mutated it to arginine, with the expectation that a constitutive positive charge might support polymerization at neutral pH. Indeed, we found that Ura7-H360R robustly assembles filaments at pH 7.4 which, like the wild-type protein, assemble only in the presence of substrate or product ligands (*Figure 4a*, *Figure 4—figure supplement 1*). Recombinant wildtype and both mutant proteins had similar purity and yield (*Figure 4—figure supplement 2a*). To confirm that H360R does not alter filament structure, we determined a 6.7 Å cryo-EM structure of substrate-bound Ura7, which is indistinguishable from wildtype at this resolution (*Figure 4—figure supplement 2b–f*, *Figure 4—figure supplement 3*). Thus, reversible assembly of H360R into wiltype-like filaments is pH-insensitive.

Decoupling polymerization from pH in engineered mutations provides tools to determine the functional consequences of polymerization on enzyme activity. Yeast CTPS activity has a strong intrinsic pH-dependence that peaks around pH 8.0 (*Nadkarni et al., 1995*; *Yang et al., 1994*), likely due to pH sensitivity of the active site cysteine in the glutaminase domain (*Trotta et al., 1973*). Consistent with this, Ura7 wildtype, H360A, and H360R all have low activity at pH 6.0. However, H360A had approximately threefold higher activity, indicating that unassembled CTPS maintains residual activity even at low pH (*Figure 4b*). Wild-type CTPS and mutants retain activity at pH

**Table 3.** Kinetic activity parameters for yeast CTPS and mutants.

|     | Wild type | H360A | H360R |
| --- | --- | --- | --- |
| Max | 2.68 ± 0.05 | 3.60 ± 0.08 | 1.18 ± 0.05 |
| Min | 0.00 ± 0.00 | 0.00 ± 0.00 | 0.00 ± 0.00 |
| S50 | 161.07 ± 14.24 | 160.03 ± 5.67 | 229.72 ± 9.26 |
| Hill | 1.53 ± 0.15 | 1.45 ± 0.03 | 0.80 ± 0.15 |

*Error represents standard error of the mean (SEM).

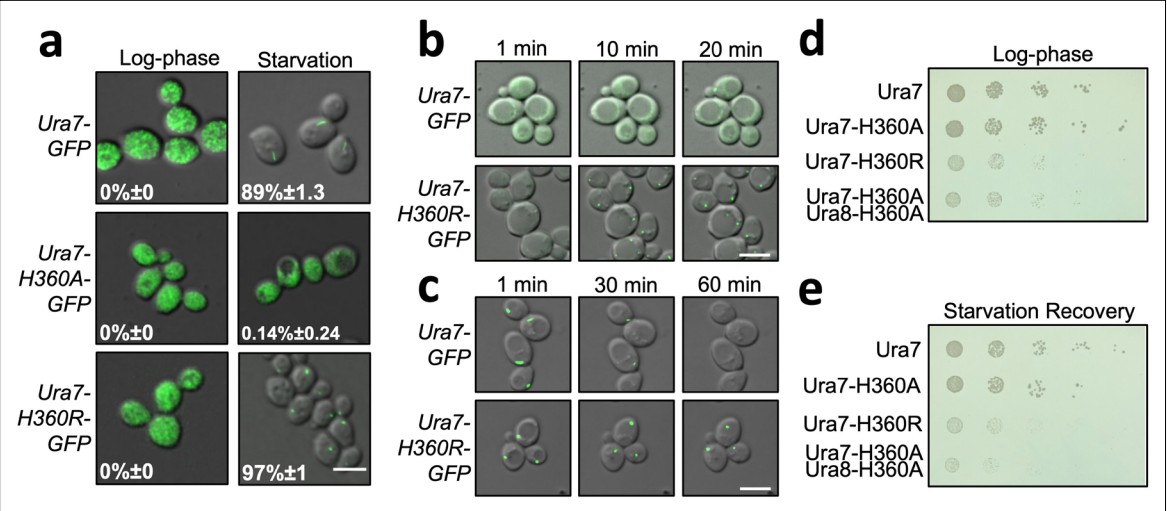

**Figure 5.** Dysregulated assembly of CTPS in yeast leads to slowed growth. (**a**) Yeast expressing GFP-tagged Ura7. Quantification is the percentage of cells showing foci. Scale bar is 5 μm. (**b**) Polymerization kinetics of GFP-tagged Ura7 in yeast upon transfer of cells to starvation media. (**c**) Same as B, for depolymerization kinetics upon re-addition of starved cells (4 hr) to nutrient-rich media. Scale bar is 5 μm. (**d**) Spot growth assay of yeast expressing wild type and mutant CTPS from liquid cultures grown in log phase. 1 in 10 dilutions increase from left to right. (**e**) Same as D, for yeast starved for 4 hr and allowed to recover on nutrient-rich solid media.

The online version of this article includes the following video and figure supplement(s) for figure 5:

**Figure supplement 1.** Ura7 assembly and disassembly kinetics.

**Figure supplement 2.** URA8 non-assembly mutant.

**Figure supplement 3.** Ura7 quantification from yeast cell lysates.

**Figure 5—video 1.** Wild-type Ura7 polymerization.

https://elifesciences.org/articles/73368/figures#fig5video1

**Figure 5—video 2.** Ura7-H360R polymerization.

https://elifesciences.org/articles/73368/figures#fig5video2

**Figure 5—video 3.** Wild-type Ura7 depolymerization.

https://elifesciences.org/articles/73368/figures#fig5video3

**Figure 5—video 4.** Wild-type Ura7-H360R depolymerization.

https://elifesciences.org/articles/73368/figures#fig5video4

7.4, where we measured the substrate kinetics of all three. The $K_M$ values were similar for wildtype and both mutants, indicating no change in affinity for the substrate UTP. However, $V_{max}$ varied inversely with the degree of filament assembly (*Figure 4c*, *Table 3*). At pH 7.4 Ura7-H360A is completely tetrameric and had the highest activity, wild-type enzyme has a background of single filaments and had slightly lower activity, and Ura7-H360R had robust filament assembly and the lowest observed activity. This suggests that filament assembly in itself acts as an allosteric inhibitor of enzyme activity, with kcat reduced likely as a consequence of the constricted ammonia channel we observe in the filament structure.

## CTPS assembly is critical for growth

To investigate the in vivo consequences of CTPS assembly, we generated yeast strains with filament assembly mutations at the endogenous locus, either with or without fluorescent protein tags, and tested both enzyme localization and cell growth (*Figure 5a*). We did not observe differences in growth between wild-type strains with and without fluorescent tags; nonetheless, the growth assays described below were all performed with untagged strains.

Based on our in vitro findings, we predicted that the pH-insensitive Ura7-H360R would constitutively assemble polymers in cells. But surprisingly, its localization was very similar to wildtype, diffuse during log phase growth and assembled into foci upon nutritional deprivation. However, Ura7-H360R-GFP assembly and disassembly kinetics are dysregulated (*Figure 5b–c*, *Figure 5—figure supplement 1*,

*Figure 5—videos 1–4*). Upon transition to starvation media, wildtype Ura7-GFP typically assembles foci over the course of 30 min, but Ura7-H360R-GFP assembles more rapidly, with virtually all cells having foci at 5 min. Even more striking, upon recovery from starvation, in which wild-type Ura7 disassembles in most cells after 30 min, at 1 hr we did not observe significant depolymerization of Ura7-H360R-GFP foci (*Figure 5—figure supplement 1*). The rapid assembly and slow disassembly of Ura7-H360R in cells are consistent with the in vitro pH-insensitive phenotype, but suggest that the starvation-triggered pH change is necessary but not sufficient to enable cellular Ura7 assembly. In addition to disrupted assembly kinetics, Ura7-H360R grew more slowly than wildtype, when plated either from log phase cultures or from starved cultures (*Figure 5d–e*), suggesting that disruption of normal Ura7 assembly and disassembly is generally detrimental to growth.

Consistent with our in vitro findings, cells expressing non-assembling Ura7-H360A-GFP or Ura8-H360A-GFP did not form foci, even under nutritional stress when the cytoplasm is acidified (*Figure 5a*). The non-assembly single mutants of Ura7 or Ura8 grew indistinguishably from wild type during log phase and upon recovery from starvation (*Figure 5d–e*, *Figure 5—figure supplement 2*). This suggested that polymerization of either CTPS isoform is sufficient to maintain normal growth. To test this, we generated the double mutant URA7-H360A/URA8-H360A, and found that it had a severe growth defect in log phase and upon starvation recovery (*Figure 5d–e*), indicating an important role for CTPS polymerization in proliferation.

We wondered whether filaments might play a role in protecting CTPS from degradation, and that perhaps the growth defect in the double H360A mutants is due to loss of the enzyme during starvation (*Petrovska et al., 2014*). To test this, we starved Ura7-GFP, Ura7-H360A-GFP, and Ura7-H360R-GFP strains and measured the concentration of CTPS in total cell lysate. Wildtype Ura7 levels are decreased during starvation, but there was no difference in protein levels between wild type and the mutants at 4 or 24 hr of starvation (*Figure 5—figure supplement 3*). This indicates that filament assembly does not play a significant role in protecting CTPS from degradation, and suggests that growth defects observed upon starvation recovery for URA7-H360A/URA8-H360A arise from other effects of defective polymerization.

It was surprising that strains with H360A or H360R mutations experienced slow growth relative to wildtype during log phase, when the mutant strains have the same diffuse localization seen in the wild-type strain. This may be consistent with the observation that wildtype enzymes have a low level of background assembly at neutral pH (*Figure 2c*), and indicates that small, transient assemblies may play a role in regulating activity during log phase growth.

## Yeast CTPS assembles large-scale ordered bundles

In wildtype cells at 30 min after nutrient deprivation, cells that had CTPS foci almost all had just a single structure (*Figure 2b*), consistent with prior reports of CTPS polymerization in yeast (*Noree et al., 2010*; *Petrovska et al., 2014*; *Shen et al., 2016*). However, during the live cell imaging experiments described above, in cells expressing wildtype Ura7-GFP we frequently observed the appearance of multiple small puncta at early time points that later coalesced into a single large structure (*Figure 6a*). This in vivo behavior is reminiscent of the rapid linear polymerization followed by lateral aggregation into larger bundles that we observed in vitro (*Figure 2—figure supplement 2*).

We determined structures of CTPS filament bundles formed at pH 6.0 to better understand the mechanisms of lateral assembly. Our initial question was whether the bundles were aggregating non-specifically, or whether there were defined assembly contacts. Reference-free two-dimensional averages of bundle segments from cryo-EM images of Ura7 and Ura8 were strikingly regular, suggesting ordered assembly contacts. This high degree of order allowed us to determine three-dimensional structures of both homologs in the presence of substrates or product using a single particle reconstruction approach by focused refinement of interacting pairs of filaments (*Figure 6b*, *Figure 6—figure supplement 1*). As we found for the single filament structures, Ura8 reconstructions went to higher resolution (3.3 Å for substrates and 3.7 Å for product) than Ura7 (6.6 Å substrate and 7.0 Å product). The individual filaments that pack laterally in the bundles closely resemble the corresponding single filament structures (*Figure 6—figure supplement 2*, *Table 2*). We observed two different bundle architectures. Ura7 formed very similar bundles in both ligand conditions, with adjacent filaments staggered relative to each other with a half-tetramer offset (*Figure 6c*). The Ura8 substrate-bound structure had the same architecture as both Ura7 bundles, but the product-bound Ura8 structure had

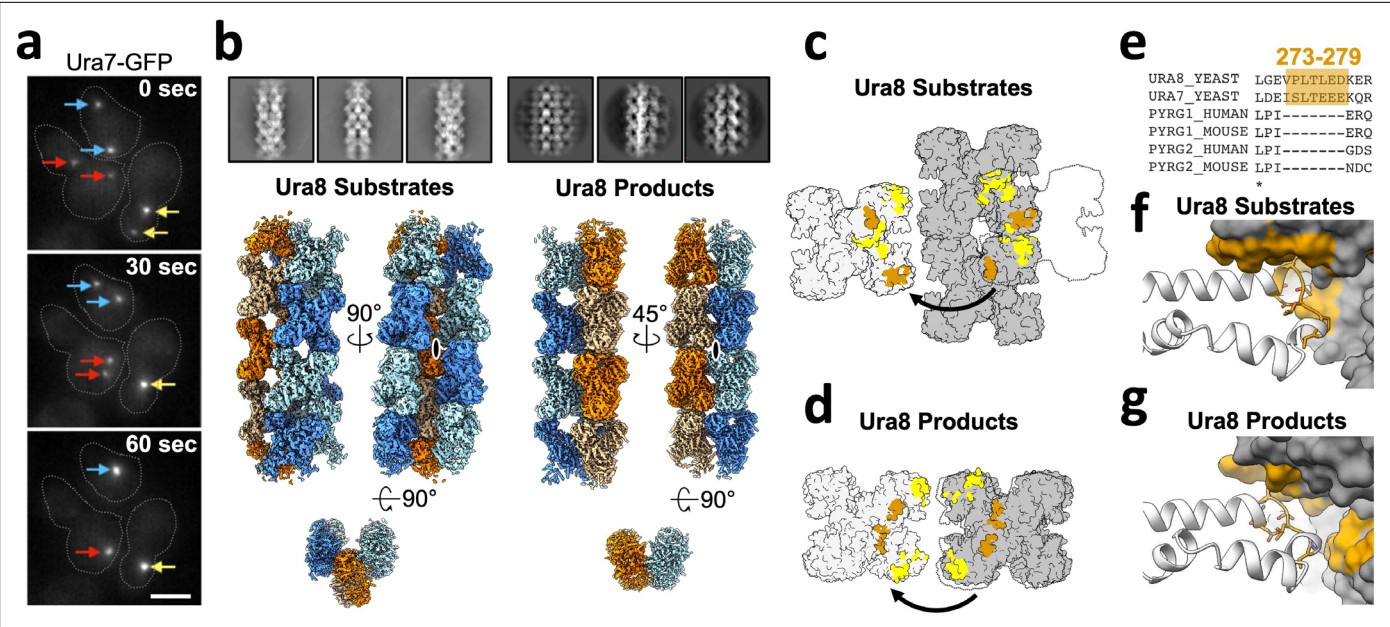

**Figure 6.** Yeast CTPS assembles highly ordered and distinct bundle architectures. (**a**) Live imaging in yeast expressing GFP-tagged Ura7. Same colored arrows indicate pairs of foci that join to larger foci at later time points. Time indicated is seconds after imaging began. Cell outlines drawn manually. Scale bar 5 μm. (**b**) Cryo-EM 2D averages and 3D reconstructions of Ura8 in the substrate (2 mM UTP/ATP) and product-bound (2 mM CTP) states. Individual strands colored in either orange or blue with protomers shaded differently. C2 symmetry axes are shown as ovals, with axis of rotation projecting toward the reader. (**c**) Lateral interactions in the substrate-bound Ura8 bundle. Full contacts are painted yellow for a single tetramer (gray on left), including redundant interactions. The yeast-specific linker insert and the interface with which it interacts are painted brown. (**d**) Same as C, for product-bound Ura8 bundle. (**e**) Sequence alignment of the CTPS linker region showing the yeast-specific insert at the specified yeast residues. (**f**) Zoom-in of the yeast linker insert of the substrate-bound bundles interacting with the adjacent strand. Numbering is the same as panel C. (**g**) Same as panel F, for product-bound bundle. Numbering is the same as panel D.

The online version of this article includes the following video and figure supplement(s) for figure 6:

**Figure supplement 1.** Ura7 bundle architecture.

**Figure supplement 2.** Filament assembly interface and nucleotides for Ura8 in bundles.

**Figure supplement 3.** Comparison of filaments from individual filaments versus strands within bundles.

**Figure supplement 4.** Numbering of monomers for assigning lateral contacts.

**Figure supplement 5.** Yeast linker involvement in lateral bundle contacts.

**Figure supplement 6.** Possible higher order arrangements of yeast CTPS bundles.

**Figure supplement 7.** Image processing of substrate-bound Ura8 bundles.

**Figure supplement 8.** Image processing of product-bound Ura8 bundles.

**Figure supplement 9.** Image processing of substrate-bound Ura7 bundles.

**Figure supplement 10.** Image processing of product-bound Ura7 bundles.

**Figure 6—video 1.** Masked cryo-EM density of substrate-bound Ura8 bundle.
https://elifesciences.org/articles/73368/figures#fig6video1

**Figure 6—video 2.** Atomic model of substrate-bound Ura8 bundle.
https://elifesciences.org/articles/73368/figures#fig6video2

**Figure 6—video 3.** Masked cryo-EM density of product-bound Ura8 bundle.
https://elifesciences.org/articles/73368/figures#fig6video3

**Figure 6—video 4.** Atomic model of product-bound Ura8 bundle.
https://elifesciences.org/articles/73368/figures#fig6video4

very different interfaces between filaments, giving rise to filaments in register (*Figure 6d*, *Figure 6— videos 1–4*).

To better understand the nature of lateral assembly contacts, we built atomic models into the two Ura8 bundle structures. Ligands were clearly visible, and bound as in the single filament

**Table 4.** CTPS bundle buried surface area and putative contact residues.

Ura8 Product-bound bundle contacts

| BSA | Domains | Monomer 0 | Monomer 1 | Monomer 2 | Monomer 3 | Monomer 4 |
|---|---|---|---|---|---|---|
| 93 Å² | Glu-Glu | 465,468,515 | 381, 384, 429,430, 432,433 | | | |
| 173 Å² | Glu-Glu | 471,472,476, 477*,480–483 | 348,350*, 352,353* | | | |
| 113 Å² | Glu-Glu | 566,567†,570,571 | 354–356,358 | | | |
| 240 Å² | Link-AL | 275,276,277†, 278,280*,281* | | 222*,225,226†, 230,236,237* | | |
| 48 Å² | Link-AL | 272,274 | | | 168,169,171,172 | |

Ura8 substrate-bound bundle contacts

| BSA | Domains | Monomer 0 | Monomer 1 | Monomer 2 | Monomer 3 | Monomer 4 |
|---|---|---|---|---|---|---|
| 471 Å² | Link-Glu | 275,276,277†, 278–282,284,285,288 | | | 348,350,380–383,384†,387,412, 416,423,425,426,429,430,432,433,435 | |
| 98 Å² | AL-Glu | 136–139 | | | 353,354 | |
| 185 Å² | Glu-AL | 471,472,477, 481,506,507,526,527 | | 168–172,174, 272,274 | | |
| 152 Å² | Glu-AL | 563,567,570,571* | | 100*, 105–107, 109,119, 123,126,127,130 | | |
| 88 Å² | Glu-AL | 465,468,471,476, 483,484 | 233,236–238,263 | | | |
| 151 Å² | AL-Glu | 168–172,174 | | | | 469,471,472,477 |
| 167 Å² | AL-Glu | 100,105–107, 123*,126†, 130* | | | | 561,563,564*, 567*,571 |

*Predicted salt bridge.

†Predicted hydrogen bond.

structures, and the longitudinal assembly interfaces were the same as observed in the single filaments (*Figure 6—figure supplement 2*). Domain rotations and the state of the closed ammonia channel were nearly identical between filaments and bundles (*Figure 6—figure supplement 3*). Unlike single Ura8 filaments determined at an intermediate pH (6.5) and with two rotamers for His360, the bundle map appeared to fit a single rotamer which points to D370 (*Figure 3—figure supplement 3d–e*). Lateral associations between filaments result in tightly packed bundles. The buried surface area per tetramer at lateral interfaces, 2202 Å² (substrates-bound) or 1676 Å² (product-bound), is comparable to the 2460 Å² involved in longitudinal filament assembly contacts, suggesting that lateral association contributes to the overall stability of the assembly (*Figure 6—figure supplement 4*, *Table 4*). Yeast CTPS has a 7-residue insert (residues 273–279) in the linker region that mediates the bulk of the lateral interaction in both bundle types (*Figure 6e*). In the substrate-bound structure, Leu277 nestles into a hydrophobic pocket formed by Ile383/Ile387/Ile412/Phe429 (*Figure 6f*, *Figure 6—figure supplement 5a,b*). In the product-bound structure the linker inserts near the tetramerization interface and packs against Pro236 and Ile225 (*Figure 6—figure supplement 5c–d*). Although at lower resolution, the Ura7 structures appear to make the same lateral contacts as substrate-bound Ura8 (*Figure 6—figure supplement 1*). Similar filament bundles have not been reported for other species, despite extensive structural characterization of human, *Drosophila*, and *E. coli* CTPS filaments (*Barry et al., 2014*; *Lynch et al., 2017*; *Lynch and Kollman, 2020*; *Zhou et al., 2021*), suggesting that the unique yeast insert promotes lateral assemblies that are specific to yeast.

Although individual filament-to-filament lateral interactions are all identical, there is variability of the assembly architecture at longer scales. The dihedral symmetry of the CTPS tetramer presents two potential lateral interfaces on each face of single filaments, but steric constraints limit occupancy to just a single lateral interaction per face. 3D classification of bundles yielded multiple structures with 3–5 associated filaments, with mixtures of cis or trans configurations of laterally associated filaments accounting for the variation (*Figure 6—figure supplement 8*). To envision potential larger bundle

architectures, we extrapolated the lateral contacts by propagating assemblies in silico (*Figure 6—figure supplement 6*). For the staggered bundle architecture, propagation of cis or trans interactions results in curved sheets. Propagation of trans interactions in the Ura8 product bound structure with filaments in register also results in a curved sheet, but propagation of cis interactions results in a closed tube with nine filaments. Mixed cis and trans interactions are also possible and observed in some of our 3-D classes, and give rise to increasingly complex structures.

## Discussion

Maintenance of balanced nucleotide pools is essential for all organisms, and CTPS plays a critical, conserved role in directly balancing pyrimidine levels. The polymerization of CTPS into cellular filamentous polymers occurs in bacteria, archaea, and in eukaryotes (*Carcamo et al., 2011*; *Ingerson-Mahar et al., 2010*; *Liu, 2010*; *Noree et al., 2010*; *Zhou et al., 2021*). Bacterial and animal CTPS filaments assemble with completely different interfaces and functional consequences for enzyme regulation, inhibiting activity in the bacterial enzymes and increasing activity or enhancing cooperative regulation in the animal enzymes (*Barry et al., 2014*; *Lynch et al., 2017*; *Lynch and Kollman, 2020*). We have shown that budding yeast CTPS assembles filaments with yet another, distinct assembly interface that maintains the enzyme in a low activity state. The diversity of CTPS filament structure and function raises the question of whether this reflects modification of an ancestral filament structure, or whether CTPS polymerization has evolved independently multiple times. There is certainly precedent for extreme divergence in the quaternary structure of filamentous polymers, a striking example being the family of actin homologs in bacteria which have maintained polymeric forms but vary in assembly contact surfaces, strand number, polarity, and handedness that give rise to functional variation (*Egelman, 2003*; *Ozyamak et al., 2013*). Independently evolved filaments, on the other hand, would be consistent with the observation that proteins with dihedral symmetry (like the CTPS tetramer) can be induced to polymerize with a very small number of mutations on their high symmetry surfaces (*Garcia-Seisdedos et al., 2017*). Given the regulatory importance of CTPS in nucleotide homeostasis, polymerization may have evolved as a relatively straightforward way to introduce a new layer of allosteric regulation to meet different demands in different lineages. Future studies of CTPS from more diverse species will likely provide greater insight into the evolution of its polymerization.

Whatever its evolutionary origins, the yeast CTPS filament has acquired features that make its assembly responsive to cytoplasmic changes in nutrient availability and growth conditions. Upon starvation-induced cytoplasmic acidification, CTPS assembly is dramatically increased in cells (*Noree et al., 2019*; *Petrovska et al., 2014*; *Shen et al., 2016*). We have shown here that yeast CTPS polymerization is a self-assembly mechanism that does not require other cellular factors, and that the pH-sensitivity of CTPS polymers is intrinsic to the enzyme itself (*Figure 2c*). The unique yeast CTPS assembly interface positions a titratable histidine residue, H360, to interact with an acidic residue, D370, an interaction that is likely strengthened at lower pH by H360 protonation (*Figure 3—figure supplement 3*). In the filament, CTPS is held in a conformation that closes off an internal ammonia channel that likely reduces activity (*Figure 3f and h*). Mutations at the assembly interface that either block assembly (H360A) or eliminate the pH sensitivity (H360R) showed that assembly is correlated with reduced activity, both at neutral pH where the wildtype enzyme forms short single polymers, and at low pH where the wild-type enzyme forms large laterally assembled bundles that are threefold less active than the non-assembling mutant (*Figure 4b*).

Lateral association of filaments into larger bundles has been observed in cells in various species by electron tomography, and suggested from fluorescence imaging (*Carcamo et al., 2011*; *Ingerson-Mahar et al., 2010*; *Noree et al., 2010*). Previous in vitro studies, however, showed purified CTPS forms primarily single filaments, and did not report extensive bundling, suggesting that the bundles observed in those cells may be induced by crowding (*Barry et al., 2014*; *Lynch et al., 2017*; *Lynch and Kollman, 2020*; *Zhou et al., 2021*). Yeast, on the other hand, has a strong propensity to assemble laterally that is intrinsic to the purified protein, driven by a yeast-specific insert in the linker domain that mediates well-ordered lateral contacts. Ura7 bundles in either ligand state are very similar, and at our resolution appear to maintain the same lateral contacts but with a slightly different helical twist. Regardless of nucleotide pools, bundles could therefore remain assembled to fulfill their role in the starvation response. Conversely, Ura8 forms bundles with radically different architectures dependent on whether substrates or products are bound; it remains unclear what the functional consequence is

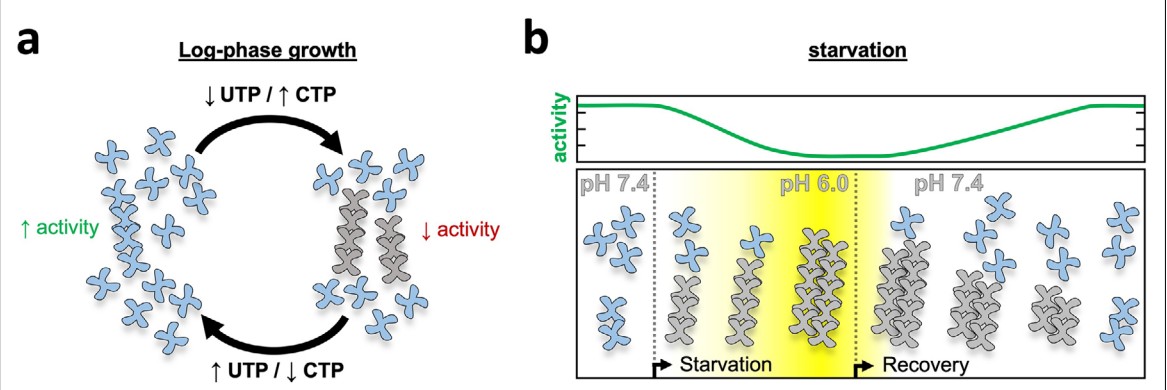

**Figure 7.** Model of yeast CTPS filament function. (**a**) Log-phase growth in yeast the majority of CTPS is in the unassembled active conformation, with a subset of enzyme polymerized to maintain nucleotide pool balance. If UTP levels are high, filaments are in the strained substrate-bound conformation which breaks assembly contacts, allowing the enzyme to remain active and process substrate while polymerized. When CTP levels are high, filaments assemble rigid product-bound filaments which inactivates the enzyme until equilibrium is reached and CTPS return to the substrate-bound filament state. (**b**) During log growth there is an equilibrium of CTPS tetramers and filaments (left). Starvation induces a slow cytoplasmic acidification which, in conjunction with a licensing event, leads to CTPS filament and bundle growth. Bundles maintain a low activity state by stabilizing filaments through lateral assembly. As yeast recover from starvation the cytoplasm is rapidly alkalinized, but bundles persist as they slowly disassemble and gradually ramp up activity toward log phase growth.

of having two distinct bundle architectures. Both types of bundles are able to accommodate lateral addition of strands in various ways, and this heterogeneity suggests that the specific architecture may be of less importance than simply maintaining protomers in a polymer.

The CTPS reaction mechanism is intrinsically sensitive to pH, with reduced activity at lower pH (*Nadkarni et al., 1995*; *Yang et al., 1994*). What then is the utility of having a redundant allosteric mechanism for inhibition by filament assembly at low pH? One possibility may be that the low residual activity of tetrameric CTPS at low pH is sufficient to imbalance nucleotide pools over prolonged starvation. Another explanation may lie in the kinetics of reactivation. Upon reintroduction of glucose the cytoplasm returns to neutral pH over about 2 min (*Orij et al., 2009*), but CTPS filaments disassemble over about 30 min or longer (*Figure 5c*, *Figure 5—figure supplement 1*), which may enable a more controlled ramping up of enzyme activity upon re-entering growth. In both our non-assembling and pH-insensitive mutants this process of controlled disassembly is disrupted, resulting in severe growth defects.

Our in vitro results correlating pH, filament assembly, and activity levels suggest a model for cellular function for CTPS filaments. Under growth conditions near neutral pH CTPS filaments modulate activity through an equilibrium of unassembled tetramers and dynamic short filaments that would be responsive to transient changes in substrate and product pools, as has been proposed for CTPS filaments of other species (*Barry et al., 2014*; *Lynch et al., 2017*). Sensitivity of CTPS polymers to the balance of substrates and products modulates transient spikes in UTP or CTP concentrations by driving assembly of inactive filaments or disassembly into active tetramers and loosely tethered flexible filaments (*Figure 7a*). Upon starvation, however, acidification acts as a signal to strongly drive assembly and inactivate the entire pool of CTPS in preparation for entering quiescence. Continued acidification drives filament growth as enzymatic activity ramps down, and eventually association of filaments into bundles (*Figure 7b* center) and lateral interactions maintain filaments in an assembled, low activity state. Upon nutrient re-addition and rapid re-alkalinization of cytoplasm, the size of bundles leads to a lag in depolymerization and a gradual ramp up of activity toward log-phase growth (*Figure 7b* right).

Disruption of polymerization-based allosteric regulation of CTPS likely results in disruption of nucleotide homeostasis, with potential consequences for processes that depend on nucleotide pools like ribosome biosynthesis, DNA replication, and phospholipid biosynthesis (*Chang and Carman, 2008*; *Fairbanks et al., 1995*). We predict that non-assembling mutants would overproduce CTP and deplete the substrate UTP, while pH-insensitive mutants would decrease CTP production, with cascading consequences for connected metabolic pathways. Further, disruption of CTPS increases

genomic instability (*Whelan et al., 1993*), and specifically disrupting Ura7 in yeast is strongly muta-genic, particularly during the stress response (*Schmidt et al., 2017*). Our characterization of these mutants lays the groundwork for future studies to examine the role of CTPS assembly in maintaining global metabolite levels and flux in pyrimidine biosynthesis.

The assembly defective mutants had localization patterns in cells that were consistent with their in vitro phenotype (*Figure 5a*). Ura7-H360A-GFP did not form large-scale cellular assemblies under any growth conditions, confirming that the filament interface we observe in cryo-EM structures is important for assembly in vivo. Ura7-H360R-GFP, on the other hand, did not assemble large structures during log phase growth, as we would have predicted based on our observation of in vitro assembly at neutral pH (*Figure 5a*). One explanation may be that cytoplasmic concentrations of ligands, which are necessary for filament assembly in this mutant, are limiting under these normal growth conditions (*Figure 4—figure supplement 2*). Alternatively, in the cellular context some factor in addition to pH may prevent assembly during log phase growth, and once this block is lifted upon nutrient deprivation the Ura7-H360R-GFP polymer rapidly assembles even before the cytoplasm has acidified. Such a licensing mechanism might prevent small oscillations in nutrient availability from triggering massive CTPS assembly by transient pH changes, but only allow assembly under more pronounced stress. One candidate for such a licensing event could be phosphorylation or dephosphorylation of CTPS. Several functionally important phosphorylation sites have been identified in yeast CTPS (*Choi et al., 2003*; *Park et al., 2003*; *Park et al., 1999*), one of which is near a bundle assembly contact (S354 in Ura7). However, there was a striking difference in in vivo assembly kinetics of Ura7-H360R-GFP which, unlike the wild-type enzyme, assembled much more rapidly than the approximately 30–60 min it takes for cytoplasmic acidification upon nutrient deprivation (*Orij et al., 2009*; *Figure 5b*, *Figure 6—figure supplement 1*).

Our findings do not rule out other functions for CTPS filaments beyond a role in allosteric enzyme regulation. In particular, one possibility is that filament formation by multiple yeast enzymes during starvation changes physical properties of the cytoplasm to a more protective solid-like state (*Petrovska et al., 2014*). CTPS filaments likely do contribute to bulk cytoplasmic changes, but our observation that the disruption of only this filament is sufficient to drastically disrupt normal growth suggests that CTPS polymers play a more specific role in managing nutrient stress. CTPS filaments may also serve other functions in scaffolding or signaling; Ura7 co-localizes with other metabolic filaments in yeast (*Noree et al., 2019*; *Noree et al., 2010*), raising the possibility that filaments provide a mechanism for direct physical interaction of enzymes for coordinated regulation of different pathways. Alternatively, bundles may serve as a signaling mechanism of nutrient deprivation for interacting partner proteins. Previous studies have identified eif2 translation initiation factor interaction with CTPS filaments in *Drosophila*, which may have direct consequences on growth through organizing protein expression (*Zhang et al., 2021*). Future studies looking at the co-assembly state of these enzymes in the context of CTPS assembly mutants will be informative for determining whether CTPS bundles play an additional role in the stress response.

## Materials and methods

**Key resources table**

| Reagent type (species) or resource | Designation | Source or reference | Identifiers | Additional information |
| --- | --- | --- | --- | --- |
| Chemical compound, drug | IPTG | GoldBio | I2481C100 | |
| Chemical compound, drug | MgCl2 | Fisher Scientific | BP215-500 | |
| Chemical compound, drug | NaCl | Fisher Scientific | S271-10 | |
| Chemical compound, drug | KCl | Fisher Scientific | BP217-3 | |
| Chemical compound, drug | Imidazole | Sigma Aldrich | SLBT7469 | |
| Chemical compound, drug | HEPES | Fisher Scientific | BP310-1 | |
| Chemical compound, drug | MES | Sigma Aldrich | M8250-100G | |
| Chemical compound, drug | TrisHCl | Fisher Scientific | BP152-5 | |

*Continued on next page*

*Continued*

| Reagent type (species) or resource | Designation | Source or reference | Identifiers | Additional information |
|---|---|---|---|---|
| Chemical compound, drug | ATP | Sigma Aldrich | A2383-10G | |
| Chemical compound, drug | UTP | Sigma Aldrich | U6750-250mg | |
| Chemical compound, drug | CTP | Sigma Aldrich | C1506-250MG | |
| Chemical compound, drug | GTP | Sigma Aldrich | G8877-1G | |
| Chemical compound, drug | Uranyl Formate | Electron Microscopy Sciences | 22,450 | Negative stain EM |
| Chemical compound, drug | Kanamycin Sulfate | Acros Organics | 450810500 | |
| Chemical compound, drug | PMSF | Sigma Aldrich | 329-98-6 | |
| Chemical compound, drug | Glycerol | Fisher Scientific | G33-1 | |
| Chemical compound, drug | glutamine | Fisher Scientific | BP379-100 | |
| Chemical compound, drug | Dithiothreitol (DTT) | Fisher Scientific | 172–25 | |
| Chemical compound, drug | 2,4-Dinitrophenol | Sigma | 51-28-5 | |
| Recombinant DNA reagent | pet28b-6His | Addgene | 73,018 | |
| Strain, strain background (*Escherichia coli*) | TOP10 | Thermo Scientific | C404003 | |
| Strain, strain background (*Escherichia coli*) | *BL21(DE3) RIL* | Thermo Scientific | EC0114 | |
| Strain, strain background (*Saccharomyces cerevisiae*) | W303 | in house | leu2-3,112 trp1-1 can1-100 ura3-1 ade2-1 his3-11,15 | |
| Other | C-flat 2/2 holey carbon film | Protochips | CF-2/2–2 C | |
| Other | Superdex 200 increase 10/300 gl hi load | GE | 28-9893-35 | Gel filtration |
| Other | Amicon Ulta-15 30 K MWCO centrifugal filters | Millipore | UFC903008 | Protein concentrator |
| Other | 5 ml HisTrap FF Crude column (GE) | GE | 17528601 | Affinity tag purification |
| Software, Algorithm | MotionCor2 | https://doi.org/10.1038/nmeth.4193 | | *Zheng et al., 2017* |
| Software, Algorithm | PHENIX | https://doi.org/10.1107/9780955360206000865 | | *Adams et al., 2012* |
| Software, Algorithm | CTFFIND4 | https://doi.org/10.1016/j.jsb.2015.08.008 | | *Rohou and Grigorieff, 2015* |
| Software, Algorithm | gCTF | https://doi.org/10.1016/j.jsb.2015.11.003 | | *Zhang, 2016* |
| Software, Algorithm | crYOLO | https://doi.org/10.1038/s42003-019-0437-z | | *Wagner et al., 2019* |
| Software, Algorithm | Relion | https://doi.org/10.7554/eLife.42166 | | *Scheres, 2012* |
| Software, Algorithm | cisTEM | https://doi.org/10.7554/eLife.35383 | | *Grant et al., 2018* |
| Software, Algorithm | cryosparc | https://doi.org/10.1038/nmeth.4169 | | *Punjani et al., 2017* |
| Software, Algorithm | Isolde | https://doi.org/10.1107/S2059798318002425 | | *Croll, 2018* |
| Software, Algorithm | Coot | https://doi.org/10.1107/S0907444910007493 | | *Emsley and Cowtan, 2004* |
| Software, Algorithm | RosettaES | https://doi.org/10.1038/nmeth.4340 | | *Frenz et al., 2017* |
| Software, Algorithm | UCSF Chimera | https://doi.org/10.1002/jcc.20084 | | *Pettersen et al., 2004* |

| Reagent type (species) or resource | Designation | Source or reference | Identifiers | Additional information |
|---|---|---|---|---|
| Software, Algorithm | Molprobity | https://doi.org/10.1107/S0907444909042073 | | *Chen et al., 2010* |

## Purification of recombinant CTPS and mutagenesis

URA7 and URA8 wild type genes with ribosomal binding site were cloned into pet28b-6His (Addgene, Massachusetts; Kan resistance, C-terminal 6 X his tags) at XhoI and XbaI sites. Mutants were generated by designing semi-overlapping primers following quickchange guidelines which incorporated the mutation (*Liu and Naismith, 2008*). Plasmids were transformed into *E. coli* BL21 (DE3) RIL cells for expression. Cultures were grown in Luria broth (LB) at 37 °C until reaching $OD_{600}$ of 0.8, then temperature reduced to 18°C for induction with 1 mM IPTG overnight. The next morning, cultures were pelleted and either stored at –80°C or protein purification carried out immediately. All subsequent steps were carried out on ice or in a 4°C cold room. Pellets from 2 L of culture were resuspended in 40 mL ice cold lysis buffer (50 mM HEPES, 1 M NaCl, 20 mM Imidizole, 10 % glycerol, 1 mM PMSF, pH 7.8) and lysed with an Emulsiflex-05 homogenizer (Avestin, Ottawa, Canada) for approximately 5 min at 15,000 PSI. Lysate was then cleared by centrifugation at 33,764 g for 30 min at 4 °C in a Thermo Scientific Fiberlite F14–14 × 50 cy rotor. Clarified lysate was loaded onto a 5 ml HisTrap FF Crude column (GE, Massachusetts) on an ÄKTA start chromatography system (GE) pre-equilibrated with lysis buffer. Unbound material was washed away with 15 column volumes of lysis buffer before an isocratic elution with five column volumes of elution buffer (50 mM HEPES, 1 M NaCl, 250 mM Imidizole, 30 % glycerol, pH 7.8). Peak fractions were combined and concentrated approximately 3-fold using a 30 kDa cut-off Amicon centrifugal filter unit (Millipore, Massachusetts). Approximately 5 mL of concentrated protein was subjected to size-exclusion chromatography using the Äkta Pure system and a Superdex 200 increase 10/300 gl pre-equilibrated with running buffer (50 mM HEPES, 1 M NaCl, 10 % glycerol, pH 7.8). Peak fractions were collected and glycerol added to a final concentration of 30 % before again concentrating using a 30 kDa cut-off Amicon centrifugal filter unit. Protein at ~2 mg/ml was flash frozen in liquid nitrogen and stored at –80 °C. Recombinant human CTPS2 was purified as described in *Lynch et al., 2020*.

## CTPS activity assays

To reduce variation between wild type and mutants (Ura7 WT, H360A and H360R), all three were purified in tandem and flash frozen. Aliquots were thawed, desalted into glycerol-free buffer (50 mM TrisHCl pH 7.4, 200 mM NaCl, 10 mM $MgCl_2$), and concentrations measured at $A_{280}$. Reactions were set up in 96 well clear plastic flat bottom corning plates (Corning, New York) with a 100 µl final reaction volume. For low pH kinetics, 2 µM protein was incubated with activity buffer (50 mM MES pH 6.0, 200 mM NaCl, 10 mM MgCl2, 10 mM BME) and nucleotide (2 mM ATP, 2 mM UTP, 0.2 mM GTP) for 15 min at 30 °C. Reaction was initiated by addition of 10 mM glutamine and absorbance measured at 291 nm in a plate reader (Varioskan lux) set at 30 °C. Early activity traces were noisy, likely due to bundled CTPS, so slope was measured between 350 and 500 s where rate of activity stabilized. A linear regression was fit to determine slope, which was used to determine CTP concentration. Replicates were performed in triplicate and averaged. Substrate kinetics were run under less optimal conditions in order to capture rates at lowest substrate concentrations. Ura7 at a final concentration of 1.5 µM was added to reaction buffer (50 mM TrisHCl 7.4, 200 mM NaCl, 10 mM MgCl2) and nucleotides (1 mM ATP, 20 µM GTP, with a range of 20µM to 1000 µM UTP). Mixture was incubated at room temperature for 5 min then inserted into a plate reader set to 22 °C and allowed to equilibrate for an additional 5–8 min, while taking readings at 291 nm until readouts were stable. The plate was briefly ejected from the reader to manually initiate the reaction by addition of glutamine (1 mM final concentration, prepared in 20 mM TrisHCl pH 7.0). CTP production was measured at 291 nm for 5 min, capturing the early linear phase of the curve. Assays were performed in triplicate then averaged, and kinetics data were fit by four parameter logistic regression $y = min + ((min-max)/(1+((max/S50)^n)))$, solving for maximum rate, minimum rate, S50, and hill number using the solver plugin in Microsoft Excel 16.51.

## Right angle light scattering

Frozen CTPS was thawed and desalted into a glycerol-free buffer (200 mM NaCl, 20 mM HEPES 7.8). Sample was set up in a total volume of 120 µl for assembly at a final protein concentration of 2 µM with 1 mM CTP. Assembly buffer was 50 mM MES 6.0 or 50 mM TrisHCl 7.0–7.4, 500 mM NaCl, and 10 mM MgCl2. Protein was added to the buffer in a quartz cuvette and inserted into a Horiba Fluorolog three fluorometer (Horiba, Kyoto, Japan) set to 30 °C and allowed to incubate until signal stabilized (approximately 3–5 min). Assembly was initiated by addition of nucleotide then readings began immediately, usually with a 5-second lag between addition and commencing readings. A total of 350 nm excitation wavelength was used and emission spectra from 350 nm were collected with 0.65 nm slit width for both. Raw data was normalized by subtracting baseline signal from initial incubation step. Sample was run in triplicate for each condition, then averaged and plotted with standard deviation in microsoft excel 16.51. Curves in *Figure 2—figure supplement 2* were fit using the four parameter logistic regression $y = min + ((min-max)/(1+(max/S50)^n))$ solving for maximum rate, minimum rate, hill number, and S50. $R^2$ values were determined using the built-in Pearson function tool in Microsoft Excel 16.51.

## Negative stain electron microscopy

CTPS was assembled in reaction buffer (50 mM TrisHCl or MES, 100 mM NaCl, 10 mM MgCl2) at 30 °C for 15 min prior to applying 5 µl of sample to glow-discharged carbon-coated grids and incubated for 1 min at room temperature. Grids were washed 3 x in $H_2O$ followed by 3 x in 0.7 % uranyl formate with blotting in between all steps to remove excess liquid. Imaging was done on an FEI Morgagni electron microscope operating at an accelerating voltage of 100 kV. Datasets for hCTPS2 were collected on a Tecnai G2 Spirit (FEI) operating at 120 kV. Images were acquired at ×67,000 magnification (pixel size 1.6 Å/px) on a Ultrascan 4000 4k × 4k CCD camera (Gatan). Contrast transfer function (CTF) was estimated using CTFFIND4 (*Rohou and Grigorieff, 2015*), Cryosparc v2.1 (*Punjani et al., 2017*) was used for automatic blob picking and 2D classification.

## Cryo-electron microscopy sample preparation and data acquisition

Frozen URA7/URA8 was desalted into minimal buffer (5 mM MES 5.9, 50 mM NaCl) prior to assembly at 8 µM with either substrates (2 µM UTP, 2 µM ATP) or product (2 µM CTP). Reaction buffer contained 400 mM NaCl, 5 mM MgCl2, 3 mM DTT, and 50 mM MES either at pH 6.0 (predominantly bundles) or pH 6.5 (predominantly single filaments). Protein was assembled for 15 minutes at 30 °C then 3 µl of the reaction material was double blotted (*Snijder et al., 2017*) onto glow-discharged C-FLAT 2/2 holey-carbon grids (Protochips), allowing a 1 min room temperature incubation between sample depositions. Grids were subsequently blotted for 4.5 s using a Vitrobot MarkIV (ThermoFisher Scientific) with chamber conditions set to 100 % humidity and at 4 °C. For Ura7 H360R sample was assembled at 5 µM final concentration in 50 mM NaCl, 50 mM HEPES 7.5, and 10 mM MgCl2 then 3 µl deposited onto lacey carbon grids which had thin carbon floated on top. Data were acquired using an FEI Titan Krios transmission electron microscope operating at 300kV and equipped with a Bioquantum GIF energy filter (Gatan) set to zero-loss mode with a slit width of 20 eV. Movies were collected on a K2 Summit Direct Detector camera (Gatan) in super-resolution mode at a magnification of 130 K (pixel size 0.525 Å/px). Automatic data acquisition was done using the Leginon Software Package (*Suloway et al., 2005*) with a defocus range specified in *Table 1*. Movies were acquired containing 50 frames having an exposure rate of 8.9 $e^-/Å^2/s$ and a total dose of 89$e^-/Å^2$. For product-bound URA8, which exhibited a severe preferred orientation, we collected data with a stage tilt of both 20 and 40 degrees and combined the data for processing.

## Cryo-EM data processing

Assembly conditions for filaments also contained some bundles, and vice versa, therefore these datasets were combined early on and bundles/filaments were separated through 2D classification and processed independently afterwards. See supplemental methods for further details. Briefly, images were manually curated to remove poor quality acquisitions such as bad ice or large regions of carbon. Dose-weighting and image alignment of all 50 frames was carried out using MotionCor2 (*Zheng et al., 2017*) with binning by a factor of 2 (final pixel size 1.05 Å/px). Initial CTF parameters were estimated using GCTF (*Zhang, 2016*). Particle picking for bundles was done using the crYOLO (*Wagner et al., 2019*) helical pickier tool trained on both filaments and bundles and using a box size covering

one tetramer, followed by manual picking to further improve the quality of the particle picks. Helical picking was chosen for the ease at the particle picking step, despite not processing any of the data with helical refinement. For tilted data, we used the local GCTF per particle CTF estimation tool to improve per particle defocus values. To classify bundles from filaments, particles were extracted with a large box size (512 pixels). Particle stacks were exported to either cisTEM (*Grant et al., 2018*) or Cryosparc (*Punjani et al., 2017*) for iterative 2D reference-free classification. Starting models for all maps were always obtained ab initio, and for bundle data the process was repeated in cisTEM and later cryosparc as independent validation. All four bundle starting models were obtained in this way, and no symmetry was imposed. 3D classification of bundles in C1 yielded subsets differing in their arrangement of strands. Classes sharing a common core assembly pattern were combined for further processing. In all cases, the highest resolution bundle maps were obtained by C2 refinement of a masked central segment after performing signal subtraction of density outside this region. Early processing without imposing symmetry suggested overall twofold symmetry in both bundle types. Therefore, we used a mask for focused alignment on three-strands (staggered bundle; Ura7 product- and substrate-bound & Ura8 substrate-bound) or two strands (in-register bundle; Ura8 product-bound) as these contained all the relevant contacts. FSC were calculated using Relion (*Scheres, 2012*) post-process or from the Phenix density modification (*Terwilliger et al., 2020*) output. Directional FSC was calculated using online FSC calculator (https://3dfsc.salk.edu/).

## Atomic model building and refinement

Initial models for Ura7 and Ura8 were obtained by threading their sequences onto the hCTPS1 substrate-bound structure from *Lynch et al., 2017*. Where inserts existed models were built manually using Coot (*Emsley and Cowtan, 2004*). Models for individual monomers were rigid body fit using Chimera (*Pettersen et al., 2004*) by domain (0–280 amido-ligase domain; 281–300 linker; 301–570 glutaminase domain) then backbone and side-chain positions refined with ISOLDE (*Croll, 2018*). C-termini were built manually in coot using the best available Ura7 map (product-bound) and Ura8 map (substrate-bound). ATP was modeled using ISOLDE, and UTP/CTP using Coot after all ISOLDE refinements. Ligands for product-bound Ura8 were rigid body fit in coot. The density corresponding to Ura8 residues 417–457 was found to be weaker and have relatively lower resolution than the rest of the structure. To build this loop the RosettaES (*Frenz et al., 2017*) loop modeling protocol in Rosetta (*Alford et al., 2017*) was used with a beamwidth of 256. The top scoring result of the RosettaES protocol was selected and its geometries were refined with ISOLDE. Residues 444–455 were later removed due to poor angles and unsupported map density. After building the monomer, it was replicated at all symmetry equivalent positions to create the full tetramer, and a single monomer at the assembly interface. Residues at the tetramerization interface and assembly interface were relaxed using a full simulation in ISOLDE. Side chains and backbone angles were adjusted for a single monomer, which was again replicated to all four symmetry equivalent sites to generate the final tetramer model with identical subunits. Models were built into the high resolution product-bound Ura7 map, which was rigid body fit by domains (residues 0–273,276-299,302-C-ter) into lower resolution Ura7 product-bound bundle map based on local map quality. Junctions between domains were deleted (residues 274–275,300-301). For substrate-bound Ura7 filament, the Ura7-product-bound filament model was rigid body fit by domain and junctions deleted (domains 0–280,281-300,301-Cter; deleted 280–284 and 300–302). Clashes for this Ura7-substrate-bound model were removed using ISOLDE. Substrate-bound Ura7 bundle was built from a rigid body fit of the substrate-bound Ura7 filament model as described above. Ura8 bundle model building began with models from their corresponding ligand-state filament, rigid body fit by domain, then refined in ISOLDE. This monomer was duplicated at all relevant contact sites and overall relaxed in ISOLDE to refine positions at tetramerization interface, filament assembly interface, and lateral bundle contacts simultaneously. The monomer was then replicated at all symmetry sites and because not each monomer experiences the same lateral contacts, it was docked into different spots and an ISOLDE simulation was run with only the residues at the lateral interface with restraints on everything else. The final monomer was duplicated into all sites to fill the map density and ligands docked in and refined using coot. Model statistics were assessed using molprobity online server () and the RCSB PDB Submission validation report. Buried surface area for lateral bundle contacts and filament interface were calculated using PDBePISA tool (https://www.ebi.ac.uk/msd-srv/prot_int/cgi-bin/piserver) with default parameters.

**Table 5.** Yeast Strains Used.

| Number | Background | Genotype |
|--------|-----------|----------|
| yJMK001 | W303 | URA7-GFP::KanMx |
| yJMK002 | W303 | URA7H360A |
| yJMK003 | W303 | URA7H360A-GFP::KanMx |
| yJMK004 | W303 | URA7H360R |
| yJMK005 | W303 | URA7H360R-GFP::KanMx |
| yJMK006 | W303 | URA8-GFP::KanMx |
| yJMK007 | W303 | URA8-mCherry::KanMx |
| yJMK008 | W303 | URA8H360A |
| yJMK009 | W303 | URA8H360A-GFP::KanMx |

## Substrate tunnel analysis

Caver 3.0.3 plugin (*Chovancova et al., 2012*) was used for Pymol 2.4.1 (*Schrödinger, LLC, 2020*) with default parameters. We set catalytic Cys404 as the start point for tunnel search of an individual monomer. Probe radius was set at 0.7 Å and the tunnel from Cys404 to the UTP base was identified. Inspection of radii along the length of the tunnel revealed that radii near the P52/C58/H55 constriction point in the Ura8 substrate-bound tetramer was between 1.2 Å and 1.9 Å, consistent with relieved ammonia channel constriction (*Lynch and Kollman, 2020*).

## Yeast strain construction and media

Yeast were maintained in standard YPD (Lab Express) or synthetic media containing 2 % D-Glucose (Fisher). Background strains were W303 *{leu2-3,112 trp1-1 can1-100 ura3-1 ade2-1 his3-11,15}*. Deletion and mutation strains were made by a PCR approach as described previously (*Wendland, 2003*), and C-terminal tagging of yeast were made as described previously (*Sheff and Thorn, 2004*). List of yeast strains used can be found in *Table 5*.

### Handling of yeast cells

To induce polymers, yeast were grown in YPD until mid-late log phase, washed in 1 X PBS and resuspended in starvation media- 0.1 M Phosphate-citrate buffer (pH 5,6,7) and grown shaking at 30 °C for 3–4 hrs. To manipulate intracellular pH in the presence of 2 % Glucose, 2 mM of 2,4-Dinitrophenol (Sigma) was added to the media as described previously (*Petrovska et al., 2014*).

### Yeast growth assays

Liquid growth curves were made by diluting mid-late log phase and/or starved yeast cells to OD600 0.05 in YPD or SD +0.5 % Glucose and growth was monitored on a Varioskan Lux plate reader at 600 nm, shaking at 30 °C for 16 hr. Solid growth assays were done by making five 5-fold serial dilutions of mid-late log phase or staved yeast cells and plated on YPD or SD +0.25 % Glucose plates +2 % Agar and grown at 30 degrees for 48 hr.

### Fluorescence microscopy

Fixed and live fluorescence microscopy was done at ×100 objective magnification on a DeltaVision Elite microscope (GE) equipped with DIC optics, using a 60 × 1.42 NA objective, and a sCMOS 5.4 PCIe air-cooled camera (PCO-TECH). Deconvolution was performed with SoftWorx (API, Issaquah, WA) and images were analyzed using Fiji, ImageJ (*Scheres, 2012*). Figures were assembled using Adobe Photoshop. Representative images shown of experiments done on three independent replicates. Fluorophores used were GFP and mCherry.

## Acknowledgements

The authors thank the Arnold and Mabel Beckman CryoEM Center at the University of Washington for electron microscope use. We thank Kelli L Hvorecny, Anika L Burrell, and John Calise for valuable feedback, especially Anika L Burrell who provided generous feedback to significantly improve the manuscript. This work was funded by the US National Institutes of Health (R01 GM118396, S10 OD032290, to JMK and T32 GM007270 to JMH). JMH thanks Kim Hansen for support and valuable discussion, and Murphy Hansen for tools aiding in structure determination.

## Additional information

### Competing interests

### Funding

| Funder | Grant reference number | Author |
|---|---|---|
| National Institutes of Health | R01 GM118396 | Justin M Kollman |
| National Institutes of Health | T32 GM007270 | Jesse M Hansen |
| National Institutes of Health | S10 OD032290 | Jesse M Hansen |

The funders had no role in study design, data collection and interpretation, or the decision to submit the work for publication.

### Author contributions
Jesse M Hansen, Conceptualization, Formal analysis, Funding acquisition, Investigation, Methodology, Validation, Visualization, Writing - original draft, Writing - review and editing; Avital Horowitz, Conceptualization, Data curation, Formal analysis, Investigation; Eric M Lynch, Conceptualization, Investigation; Daniel P Farrell, Investigation; Joel Quispe, Methodology; Frank DiMaio, Software, Supervision; Justin M Kollman, Conceptualization, Formal analysis, Funding acquisition, Methodology, Supervision, Writing - original draft, Writing - review and editing

### Author ORCIDs
Jesse M Hansen ![ORCID] http://orcid.org/0000-0001-7967-2085
Eric M Lynch ![ORCID] http://orcid.org/0000-0001-5897-5167
Frank DiMaio ![ORCID] http://orcid.org/0000-0002-7524-8938
Justin M Kollman ![ORCID] http://orcid.org/0000-0002-0350-5827

### Decision letter and Author response
Decision letter https://doi.org/10.7554/eLife.73368.sa1
Author response https://doi.org/10.7554/eLife.73368.sa2

## Additional files

### Supplementary files
• Transparent reporting form

### Data availability
Models deposited to PDB as: 7RL0, 7RNR, 7RKH, 7RL5, 7RNL, 7RMF, 7RMK, 7RMC, 7RMO, 7RMV. Maps deposited to EMDB as: EMD-24512, EMD-24581, EMD-24497, EMD-24516, EMD-24579, EMD-24566, EMD-24575, EMD-24560, EMD-24576, EMD-24578.

The following dataset was generated:

| Author(s) | Year | Dataset title | Dataset URL | Database and Identifier |
|---|---|---|---|---|
| Hansen JM, Lynch EM, Farrell DP, DiMaio F, Quispe J, Kollman JM | 2021 | Yeast CTP Synthase (URA8) Filament bound to ATP/UTP at low pH | https://www.rcsb.org/structure/7RL0 | RCSB Protein Data Bank, 7RL0 |
| Hansen JM, Lynch EM, Farrell DP, DiMaio F, Quispe J, Kollman JM | 2021 | Yeast CTP Synthase (Ura8) Bundle Bound to Substrates at Low pH | https://www.rcsb.org/structure/7RNR | RCSB Protein Data Bank, 7RNR |

*Continued on next page*

*Continued*

| Author(s) | Year | Dataset title | Dataset URL | Database and Identifier |
|---|---|---|---|---|
| Hansen JM, Lynch EM, Farrell DP, DiMaio F, Quispe J, Kollman JM | 2021 | Yeast CTP Synthase (URA8) tetramer bound to ATP/UTP at neutral pH | https://www.rcsb.org/structure/7RKH | RCSB Protein Data Bank, 7RKH |
| Hansen JM, Lynch EM, Farrell DP, DiMaio F, Quispe J, Kollman JM | 2021 | Yeast CTP Synthase (URA8) filament bound to CTP at low pH | https://www.rcsb.org/structure/7RL5 | RCSB Protein Data Bank, 7RL5 |
| Hansen JM, Lynch EM, Farrell DP, DiMaio F, Quispe J, Kollman JM | 2021 | Yeast CTP Synthase (Ura7) H360R Filament bound to Substrates | https://www.rcsb.org/structure/7RNL | RCSB Protein Data Bank, 7RNL |
| Hansen JM, Lynch EM, Farrell DP, DiMaio F, Quispe J, Kollman JM | 2021 | Substrate-bound Ura7 filament at low pH | https://www.rcsb.org/structure/7RMF | RCSB Protein Data Bank, 7RMF |
| Hansen JM, Lynch EM, Farrell DP, DiMaio F, Quispe J, Kollman JM | 2021 | Yeast CTP Synthase (Ura7) Bundle bound to substrates at low pH | https://www.rcsb.org/structure/7RMK | RCSB Protein Data Bank, 7RMK |
| Hansen JM, Lynch EM, Farrell DP, DiMaio F, Quispe J, Kollman JM | 2021 | Yeast CTP Synthase (Ura7) filament bound to CTP at low pH | https://www.rcsb.org/structure/7RMC | RCSB Protein Data Bank, 7RMC |
| Hansen JM, Lynch EM, Farrell DP, DiMaio F, Quispe J, Kollman JM | 2021 | Yeast CTP Synthase (Ura7) Bundle bound to Products at low pH | https://www.rcsb.org/structure/7RMO | RCSB Protein Data Bank, 7RMO |
| Hansen JM, Lynch EM, Farrell DP, DiMaio F, Quispe J, Kollman JM | 2021 | Yeast CTP Synthase (Ura7) H360R Filament bound to Substrates | https://www.rcsb.org/structure/7RMV | RCSB Protein Data Bank, 7RMV |
| Hansen JM, Lynch EM, Farrell DP, DiMaio F, Quispe J, Kollman JM | 2021 | Yeast CTP Synthase (URA8) Filament bound to ATP/UTP at low pH | https://www.ebi.ac.uk/emdb/EMD-24512 | Electron Microscopy Data Bank, EMD-24512 |
| Hansen JM, Lynch EM, Farrell DP, DiMaio F, Quispe J, Kollman JM | 2021 | Yeast CTP Synthase (Ura8) Bundle Bound to Substrates at Low pH | https://www.ebi.ac.uk/emdb/EMD-24581 | Electron Microscopy Data Bank, EMD-24581 |
| Hansen JM, Lynch EM, Farrell DP, DiMaio F, Quispe J, Kollman JM | 2021 | Yeast CTP Synthase (URA8) tetramer bound to ATP/UTP at neutral pH | https://www.ebi.ac.uk/emdb/EMD-24497 | Electron Microscopy Data Bank, EMD-24497 |
| Hansen JM, Lynch EM, Farrell DP, DiMaio F, Quispe J, Kollman JM | 2021 | Yeast CTP Synthase (URA8) filament bound to CTP at low pH | https://www.ebi.ac.uk/emdb/EMD-24516 | Electron Microscopy Data Bank, EMD-24516 |
| Hansen JM, Lynch EM, Farrell DP, DiMaio F, Quispe J, Kollman JM | 2021 | Yeast CTP Synthase (Ura7) H360R Filament bound to Substrates | https://www.ebi.ac.uk/emdb/EMD-24579 | Electron Microscopy Data Bank, EMD-24579 |
| Hansen JM, Lynch EM, Farrell DP, DiMaio F, Quispe J, Kollman JM | 2021 | Substrate-bound Ura7 filament at low pH | https://www.ebi.ac.uk/emdb/EMD-24566 | Electron Microscopy Data Bank, EMD-24566 |

*Continued on next page*

*Continued*

| Author(s) | Year | Dataset title | Dataset URL | Database and Identifier |
|---|---|---|---|---|
| Hansen JM, Lynch EM, Farrell DP, DiMaio F, Quispe J, Kollman JM | 2021 | Yeast CTP Synthase (Ura7) Bundle bound to substrates at low pH | https://www.ebi.ac.uk/emdb/EMD-24575 | Electron Microscopy Data Bank, EMD-24575 |
| Hansen JM, Lynch EM, Farrell DP, DiMaio F, Quispe J, Kollman JM | 2021 | Yeast CTP Synthase (Ura7) filament bound to CTP at low pH | https://www.ebi.ac.uk/emdb/EMD-24560 | Electron Microscopy Data Bank, EMD-24560 |
| Hansen JM, Lynch EM, Farrell DP, DiMaio F, Quispe J, Kollman JM | 2021 | Yeast CTP Synthase (Ura7) Bundle bound to Products at low pH | https://www.ebi.ac.uk/emdb/EMD-24576 | Electron Microscopy Data Bank, EMD-24576 |
| Hansen JM, Lynch EM, Farrell DP, DiMaio F, Quispe J, Kollman JM | 2021 | Yeast CTP Synthase (Ura7) H360R Filament bound to Substrates | https://www.ebi.ac.uk/emdb/EMD-24578 | Electron Microscopy Data Bank, EMD-24578 |

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
