## [Editor Report]

This work provides valuable new information to those who study enzyme mechanisms, nucleotide metabolism, and the response of cells to stress such as nutrient deprivation. The study focuses on CTP Synthase (CTPS), an important enzyme in nucleotide biosynthesis that has been shown to assemble into foci and filaments in yeast cells undergoing starvation conditions. The authors study the structure of yeast CTPS and its propensity to polymerize in low pH (mimicking starvation conditions), and how CTPS filamentation relates to the cellular assemblies.

---

## [Decision Letter]

**Decision letter after peer review:**

Thank you for submitting your article "Cryo-EM Structures of CTP Synthase Filaments Reveal Mechanism of pH-Sensitive Assembly During Budding Yeast Starvation" for consideration by *eLife*. Your article has been reviewed by 3 peer reviewers, including Edward H Egelman as Reviewing Editor and Reviewer #1, and the evaluation has been overseen by Cynthia Wolberger as the Senior Editor.

Essential revisions:

The authors may want to consider investigating the issue of why the "constitutive" assembly mutant does not polymerize under standard conditions in vivo, as suggested, by looking at the concentration dependence in vitro.

*Reviewer #1 (Recommendations for the authors):*

The reference to Lynch et al., under review, is not a proper citation. This should either be posted as a preprint and cited properly, or removed.

*Reviewer #2 (Recommendations for the authors):*

1. "*E. coli*" and "*Drosophila*" should be italicized throughout.

2. DNT (line 117) and DNP (one 152) presumably refer to 2,4-dinitrophenol to permeabilize membranes though not all readers would recognize what this is from its initials. I recommend spelling out the name of the compound at first use, or referring to a table so that the reader can quickly understand what's being done.

3. The equations used to fit data in figure 2 supp 2 should be better described in the methods (ie the mathematical form given).

4. Figure 2e, should the y-axis be intensity, not absorbance? It's my understanding the right angle-light scattering does not measure absorbance, which would be done in a format where the detector is in-line with the light source.

5. Figure 3b, the relationship between the two views is given as 180 degrees, but only tryptophan residues are shown making it difficult to relate the two views. Perhaps the α helices could be shown in a very light color to identify the relative locations of the tryptophan side chains.

6. Typo line 414 "interacts" should presumably be "interactions".

7. Figures 6c-d are confusing.

8. Figure 7. Do the shades of blue relate to activity? If so, why is activity higher in the filaments on the left of panel a? I see that they are in the less tethered state, but is activity higher in this state? Also, the arrows associated with UTP and CTP concentrations and leading to more active and inactive seem backwards. Shouldn't higher UTP lead to more activity not less?

9. line 579, "Superdex" should be capitalized as it is a proper noun. p592, Varioskan Lux is also a proper noun.

10. line 623, should not TRIS be written as TrisHCl (or other depending on counter ion)?

11. line 843, CTPS needs to be capitalized.

12. Figure 5—figure supplement 3, panel a is confusing in the sense that the slowest running band is marked as "α-GFP", but I believe this to be Ura7-GFP, correct?

13. Is there an estimate for error on the parameters in Table 3?

*Reviewer #3 (Recommendations for the authors):*

Major suggestion to help address my major concern: In the discussion, they suggest that a nearby phosphorylation site (S354) could be responsible. So it would be nice if they mutated that. An alternative would be to explore the concentration dependence of assembly in vitro and attempt overexpression in vivo.

---

## [Author Response]

Essential revisions:The authors may want to consider investigating the issue of why the "constitutive" assembly mutant does not polymerize under standard conditions in vivo, as suggested, by looking at the concentration dependence in vitro.Reviewer #1 (Recommendations for the authors):The reference to Lynch et al., under review, is not a proper citation. This should either be posted as a preprint and cited properly, or removed.

We have updated the reference list to correctly cite this paper, which was just recently accepted for publication.

Reviewer #2 (Recommendations for the authors):1. "*E. coli*" and "*Drosophila*" should be italicized throughout.

This has been fixed throughout.

2. DNT (line 117) and DNP (one 152) presumably refer to 2,4-dinitrophenol to permeabilize membranes though not all readers would recognize what this is from its initials. I recommend spelling out the name of the compound at first use, or referring to a table so that the reader can quickly understand what's being done.

DNP has been spelled out in the first introduction of the acronym.

3. The equations used to fit data in figure 2 supp 2 should be better described in the methods (ie the mathematical form given).

The methods have been updated to include the mathematical formula used in Figure 2 Supp 2.

4. Figure 2e, should the y-axis be intensity, not absorbance? It's my understanding the right angle-light scattering does not measure absorbance, which would be done in a format where the detector is in-line with the light source.

We thank the reviewer for pointing this out, and have modified the figure accordingly.

5. Figure 3b, the relationship between the two views is given as 180 degrees, but only tryptophan residues are shown making it difficult to relate the two views. Perhaps the α helices could be shown in a very light color to identify the relative locations of the tryptophan side chains.

We have modified Figure 3b to include parts of the α helices in order to aid in orienting the reader in that panel.

6. Typo line 414 "interacts" should presumably be "interactions".

Corrected this typo.

7. Figures 6c-d are confusing.

These panels have been modified to remove the numbers, which have been instead placed as a supplemental figure (Figure 6 Supplemental 4) in order for Table 4 to remain interpretable. Figure 6c-d now shows the higher level relationship between the two strands in a way that we hope is more clear than before.

8. Figure 7. Do the shades of blue relate to activity? If so, why is activity higher in the filaments on the left of panel a? I see that they are in the less tethered state, but is activity higher in this state? Also, the arrows associated with UTP and CTP concentrations and leading to more active and inactive seem backwards. Shouldn't higher UTP lead to more activity not less?

We thank the reviewer for pointing out the incorrectly placed arrows, this has been corrected. We have also revised the color scheme in this figure to reflect more accurately the model that we propose.

9. line 579, "Superdex" should be capitalized as it is a proper noun. p592, Varioskan Lux is also a proper noun.

This has been corrected.

10. line 623, should not TRIS be written as TrisHCl (or other depending on counter ion)?

This has been corrected throughout.

11. line 843, CTPS needs to be capitalized.

This has been corrected.

12. Figure 5—figure supplement 3, panel a is confusing in the sense that the slowest running band is marked as "α-GFP", but I believe this to be Ura7-GFP, correct?

We thank the reviewer for pointing this out, it has been corrected.

13. Is there an estimate for error on the parameters in Table 3?

We have updated the table to include SEM errors of estimate.

Reviewer #3 (Recommendations for the authors):Major suggestion to help address my major concern: In the discussion, they suggest that a nearby phosphorylation site (S354) could be responsible. So it would be nice if they mutated that. An alternative would be to explore the concentration dependence of assembly in vitro and attempt overexpression in vivo.

We thank the reviewer for their suggestion. We have modified the descriptions in the paper to reflect the characterization of our mutant as pH-insensitive rather than constitutively assembled. By characterizing the filaments as “constitutive” we omitted a clearer explanation that the mutant, like the WT, still requires ligands to support filament assembly; under apo conditions the mutant does not assemble polymers (Figure 4 Supplemental 2). So, H360R experiences reversible polymerization in vitro depending on ligand conditions, and our previous characterization as constitutively polymerizing was inaccurate. We have edited the text to make this clearer, and to refer to the unassembled state in the supplement.

Additional mutations of potential phosphoregulatory sites, and overexpression studies are, we believe, beyond the scope of the current paper, but we think are excellent jumping off points for future in vivo studies.